

**Wintertime organic and inorganic aerosols in Lanzhou, China:**
**Sources, processes and comparison with the results during**
**summer**
**J. Xu[1], J. Shi[2], Q. Zhang[3], X. Ge[4], F. Canonaco[5], A. S. H. Prévôt[5, 6], M. Vonwiller[7], S.**
**Szidat[7], J. Ge[2], J. Ma[8], Y. An[1], S. Kang[1], D. Qin[1]**
[1]State Key Laboratory of Cryospheric Sciences, Cold and Arid Regions Environmental
and Engineering Research Institute, CAS, Lanzhou 730000, China
[2]Key Laboratory for Semi-Arid Climate Change of the Ministry of Education, College of
Atmospheric Sciences, Lanzhou University, Lanzhou 730000, China
[3]Department of Environmental Toxicology, University of California, Davis, CA 95616,
USA
[4]Jiangsu Key Laboratory of Atmospheric Environment Monitoring and Pollution Control
(AEMPC), School of Environmental Science and Engineering, Nanjing University of
Information Science & Technology, Nanjing 210044, China
[5]Laboratory of Atmospheric Chemistry, Paul Scherrer Institute (PSI), Villigen 5232,
Switzerland
[6]State Key Laboratory of Loess and Quaternary Geology and Key Laboratory of Aerosol
Chemistry and Physics, Institute of Earth Environment, Chinese Academy of Sciences,
710075 Xi'an, China
[7]Department of Chemistry and Biochemistry & Oeschger Centre for Climate Change
Research, University of Bern, Berne, 3012, Switzerland
[8]College of Earth Environmental Science, Lanzhou University, Lanzhou 730000, China
Correspondence to: J. Xu (jzxu@lzb.ac.cn)



**Abstract**
Lanzhou, which is located in a steep Alpine valley in western China, is one of the most
polluted cities in China during the wintertime. In this study, an Aerodyne high resolution
time-of-flight aerosol mass spectrometer (HR-ToF-AMS), a seven-wavelength
aethalometer, and a scanning mobility particle sizer (SMPS) were deployed during
January 10 to February 4, 2014 to study the mass concentrations, chemical processes, and
sources of sub-micrometer particulate matter ($PM_1$). The average $PM_1$ concentration
during this study was 57.3 μg m$^{-3}$ (ranging from 2.1 to 229.7 μg m$^{-3}$ for hourly averages)
with organic aerosol (OA) accounting for 51.2%, followed by nitrate (16.5%), sulphate
(12.5%), ammonium (10.3%), black carbon (BC, 6.4%), and chloride (3.0%). The mass
concentration of $PM_1$ during winter was more than twice the average value observed at
the same site in summer 2012 (24.5 μg m$^{-3}$), but the mass fraction of OA was similar in
the two seasons. Nitrate contributed a significantly higher fraction to the $PM_1$ mass in
winter compared to summer, largely due to more favoured partitioning to the particle
phase at low air temperature. The mass fractions of both OA and nitrate increased by ~5%
with the increase of the total $PM_1$ mass loading, while the average sulphate fraction
decreased by 6%, indicating the importance of OA and nitrate for the heavy air pollution
events in Lanzhou. The size distributions of OA, nitrate, sulphate, ammonium, and
chloride all peaked at ~500 nm with OA being slightly broader, suggesting that aerosol
particles were externally mixed during winter, likely due to stagnant air conditions
(average wind speed: 0.82 m s$^{-1}$). All species showed evident diurnal variations reflecting
the important local/regional sources.
The average mass spectrum of OA showed a medium oxidation degree (average O/C
ratio of 0.28), which was lower than that during summer 2012. This is consistent with
weaker photochemical processing during winter. Positive matrix factorization (PMF)
with the multi-linear engine (ME-2) solver identified six OA sources, i.e., a hydrocarbon-
like OA (HOA), a biomass burning OA (BBOA), a cooking-emitted OA (COA), a coal





combustion OA (CCOA), and two oxygenated OA (OOA) factors. One of the OOAs was
semi-volatile (SV-OOA) and the other one of low-volatility (LV-OOA). SV-OOA was
the most abundant OA component (24% of OA mass), followed by COA (20%), LV-
OOA (19%), CCOA (18%), BBOA (11%), and HOA (9%). The mass fraction of primary
OA (= HOA + BBOA + COA + CCOA) increased during high PM pollution periods,
indicating that local primary emissions were a main reason for the formation of haze in
Lanzhou during winter. The primary OA sources were more complex during winter than
during summer. Radiocarbon ($^{14}$C) measurement was conducted on four $PM_{2.5}$ filter
samples from this study, which allowed for a quantitative source apportionment of
organic carbon (OC). The non-fossil sources on average accounted for $55 \pm 3\%$ of OC
which could be mainly from biomass burning and cooking activities, suggesting the
importance of non-fossil sources for the PM pollution in Lanzhou. Combining with the
PMF results, we also found that a large fraction (57%) of the secondary OC was from
non-fossil OC.

**1    Introduction**
Frequent haze pollution events in urban areas in China have been a widespread concern
in recent years due to its high adverse health effects, visibility degradation and climate
effects (Chan and Yao, 2008). The Chinese Central Government had put in extensive
efforts to find urgent and suitable control strategies to reduce further deterioration of air
quality. Strategies such as promoting energy conservation and emission reduction
measures and new air quality standards ($PM_{2.5}$ currently *vs*. $PM_{10}$ in the past) have been
implemented in the last three years. Many local governments have also launched
measures such as shutting down some highly polluting factories and restricting the use of
private vehicles to reduce air pollution in their cities. However, air pollution in China is
still far from being controlled due to its complex sources and limited knowledge on the
multiple pathways leading to secondary aerosol formation and dynamic variation of
aerosol mass loading.





Lanzhou, the capital of Gansu province, is located at the northwest of China and has
experienced air pollution issues since the 1960s due to emissions from the petrochemical
industry and its valley terrain. Air pollution is still serious and has become more variable
in recent years (since 2000) because of fast urbanization and increased energy
consumption. The severity of air pollution often reaches maximum intensity during
winter due to coal combustion for domestic heating and cooking, similar to the situations
in most cities of northern China (Wang et al., 2014); In addition, the stagnant
meteorological conditions during winter in Lanzhou such as calm wind and shallow
planetary boundary layer (PBL) caused by its valley terrain, are also important factors
influencing the transport and diffusion of air pollutants. Despite the serious air pollution
during winter in Lanzhou, aerosol chemistry, sources, and formation and transformation
processes were poorly documented in the literature, which limit the development and
implementation of efficient control strategies.

The chemical and physical properties of atmospheric aerosol particles during winter,
especially during haze episode, have been investigated in metropolitan cities in Eastern
China (Sun et al., 2006; Zhao et al., 2013; Huang et al., 2014; Sun et al., 2014). For
example, the mean aerosol optical depth at 500 nm were up to ~0.7 during the month-
long heavy haze pollution episode during January 2013 in Beijing (Bi et al., 2014); The
airborne microbes were found in particulate matter (PM) during hazy period which may
potentially include respiratory microbial allergens and pathogens (Cao et al., 2014).
Collection and analysis of filter samples have enabled quantification of the chemical
composition of PM using a suite of off-line instruments (such as ion chromatography,
organic and element carbon analyzer, inductively coupled plasma-mass spectrometry and
so on) in the laboratory (He et al., 2001; Zheng et al., 2005; Sun et al., 2006; Sun et al.,
2011a; Zhang et al., 2013; Zhao et al., 2013). Semi-continuous measurement with hourly
time resolution on chemical composition of aerosol particle is better compared to filter
collection for elucidating the chemical processes during haze episodes (Gao et al., 2011;
Xue et al., 2014; Zheng et al., 2015).

Previous studies on source apportionment of aerosol particle identified dust, traffic,
industry, cooking-related activities, and secondary formation as important contributors,
although the contributions of individual sources may change drastically with location,
season, and different apportionment algorithms (Zheng et al., 2005; Yu et al., 2013;
Huang et al., 2014). For example, Zheng et al. (2005) used chemical mass balance
(CMB) to quantitatively apportion the sources that contribute to fine PM concentration in
Beijing and found coal combustion contributed 16% of fine PM mass in January. By
contrast, principal component analysis of the same dataset estimated almost twice amount
of aerosols from coal combustion (Song et al., 2006). Source apportionment techniques -
such as positive matrix factorization (PMF) allow for using thousands of individual
species for source identification and the error of analysis to constrain the fitting, and
would thus appear more suitable to identify and apportion PM to their sources. Compared
with the number of aerosol source apportionment studies using PMF in Eastern China
(e.g., Sun et al., 2013b; Zhang et al., 2013), there were fewer studies in inland cities of
China (Elser et al., 2016), the results of which can be used for inter-comparison and
understanding the difference of aerosol pollution in different parts of China. In addition,
it has been known that a large mass fraction of ambient PM during haze episodes is from
fine particles, of which secondary species (some carbonaceous components, sulphate,
nitrate, and ammonium) are major components (Zhao et al., 2013). However, the
formation and evolution mechanisms of those secondary species were relatively poorly
understood (Sun et al., 2014). The possible fast chemical reactions occurring during haze
pollution further complicate our understanding of the formation processes of secondary
aerosol, where high time resolution instruments are likely the best tools to be used.

Online instruments based on mass spectrometric techniques, such as Aerodyne aerosol
mass spectrometer (AMS) (Jayne et al., 2000), appear to be the most advanced on
probing the fast aerosol chemical processes because of the output of a large amount of
chemical information and its fine time resolution (in minutes) and mass sensitivity (in ng



m$^{-3}$) (Canagaratna et al., 2007). Aerodyne high resolution time-of-flight mass
spectrometer (HR-ToF-AMS) have been widely employed for the chemical
characterization of submicron aerosol (PM$_1$) (DeCarlo et al., 2006). The instrument
provides on-line quantitative mass spectra of the non-refractory (inorganic and organic)
aerosol components with high time resolution. Frequently, the organic aerosol (OA) can
be further analyzed using the PMF algorithm (Ulbrich et al., 2009; Zhang et al., 2011b),
which can represent the organic mass spectral matrix as a set of source/process-related
factor mass spectra and time series. In addition, carbon isotope technique has been
recently applied to quantify the fossil/non-fossil origins of carbonaceous aerosols, and in
combination with AMS-PMF analyses, the assessment of the origin of secondary organic
aerosol (SOA) became possible (Minguillon et al., 2011; Huang et al., 2014; Zotter et al.,
2014; Beekmann et al., 2015).

In a previous study, we used an HR-ToF-AMS, to investigate the chemical characteristics
PM$_1$ in the urban area of Lanzhou during summer 2012 (Xu et al., 2014). During that
study, organics in PM$_1$ was found to mainly originate from traffic, cooking activities, and
chemical reactions which produced semi-volatile and less-volatility oxygenated OA.
Compared to summer situation, energy consumption for heating is huge during winter
and the dry and stable meteorological condition in the basin leads to longer aerosol
lifetime during winter. Thus aerosols are influenced largely by very different chemical
processes between the two seasons. Thus, more intensive measurements of PM chemical
characteristics are needed to better understand aerosol sources, to quantify their lifetime
in the atmosphere and to constrain the uncertainties of their climatic influences. During
winter of 2013/2014, we conducted such a study at an urban site of Lanzhou. In this
paper, we focus on the chemical speciation of PM$_1$ and source apportionment of OA.



## 2 Measurement and methods

### 2.1 Sampling site

Aerosol particle measurements were conducted from January 10 to February 4, 2014, at the top floor of a twenty-two story building (~70 m a.g.l) (36.05 N; 103.85 W, 1569 m a.s.l) in the campus of Lanzhou University (Fig. S1a). The campus is located in the Chenguan district of Lanzhou which is a cultural and educational area. The twenty-two story building sits at the western edge of the campus and faces a south-northern arterial road (Fig. S1a). At the campus side of this building, there is a three story dining hall of Lanzhou University, and over the arterial road side, there are many restaurants and residents. The room temperature was kept at ~20 ℃ by a central heating radiator. The weather in Lanzhou during the campaign was cold (avg. T = 0.5 ℃) and dry (avg. RH = 28%), and was influenced by the Asian winter monsoon. Because Lanzhou is surrounded by mountains, atmospheric condition is normally stable with low wind speed (0.82 m s$^{-1}$ during this study). The sampling site represents a typical urban area dominated by residential and commercial area.

### 2.2 Instruments

The physiochemical properties of aerosol particles were monitored in real-time by a suite of instruments (Fig. S1b). The sampling inlet, constructed using 0.5 inch copper tube, stemmed out of the rooftop by about 1.5 m. A $PM_{2.5}$ cyclone (model URG-2000-30EH, URG Corp., Chapel Hill, NC, USA) was used for removing coarse particles. The length of the sampling line was about 5 m. A diffusion dryer was placed upstream of this line to eliminate potential relative humidity (RH) effect on particles. The inlet was shared by an Aerodyne HR-ToF-AMS (Aerodyne, Inc., Billerica, MA, USA) for the size-resolved chemical speciation of non-refractory sub-micrometer PM (NR-$PM_1$), a single particle intra-cavity laser induced incandescence photometer (SP2, DMT, Inc., Boulder, CO, USA) for  refractory black carbon (rBC) measurement, a customer-made scanning mobility particle sizer (SMPS) (Wiedensohler et al., 2012) for measuring particle size distribution between 10-800 nm, and a 7-λ aethalometer (model AE31, Magee Scientific,





Berkeley, CA, USA) to derive the mass concentration of light absorbing black carbon
(BC) particles. The total air flow rate from the inlet was ~ 16 L min$^{-1}$, with a vacuum
pump drawing the air at a flow rate of 10 L min$^{-1}$ and the other 6 L min$^{-1}$ sampled by the
instrument. The retention time of particles in the sampling line was less than 2.5 s. A
parallel inlet with a 1:10 dilution stage was setup for real-time PM$_{2.5}$ measurement using
a tapered element oscillating microbalance (TEOM series 1400a, R&P, East Greenbush,
NY, USA). The roof of the building also hosted instruments for monitoring
meteorological parameters such as visibility, air temperature, wind direction, wind speed,
and RH. The visibility was measured with a LED-based (880 nm) forward (42 °)
scattering visibility sensor (model M6000, Belfort Ins., Maryland, USA).

2.2.1   HR-ToF-AMS operation
A detailed description of the principle and design of HR-ToF-AMS can be found
elsewhere (Jayne et al., 2000; DeCarlo et al., 2006). Briefly, HR-ToF-AMS consists of
three major sections: the inlet system, the particle sizing vacuum chamber, and the
particle composition detection section. The combination of a 100 μm orifice and an
aerodynamic lens in the inlet system are used to focus the airborne particles into a
concentrated and narrow beam, and then accelerated into the vacuum chamber (~10$^{5}$
Torr) modulated by a chopper for measuring aerodynamic size of the particle; Before
being detected, the particles are flash vaporized under 600 °C and ionized by a 70 eV
electron impact, and finally detected by the high resolution time-of-flight mass
spectrometer. The chopper works at three positions alternately, i.e., an open position
which transmits the particle beam continuously, a close position which blocks the particle
beam completely, and a chopping position which modulates the beam transmission (2%
duty cycle). The open and close positions yield the bulk and background signals for the
airborne particle, respectively, while the chopping position modulates the particle beam
by spinning chopper wheel (~150 Hz) to yield size-resolved spectral signals. The mass
spectrometer in the detection section works in two modes based on the ionic path, i.e., V-
mode and W-mode, with high sensitivity and high chemical resolution (~6000 m/Δm),



respectively. The highly sensitive V-mode signals are usually used for reporting mass
concentration, while the high chemical resolution W-mode signals are used for the
analyses of mass spectrum. The time resolution for both V and W modes was 5 min.
Under V-mode, the instrument switched between the mass spectrum mode and the PToF
mode every 15 s, spending 6 and 9 s on each, and cycled 20 times in one run; No PToF
data were recorded in W-mode due to low signal-to-noise (S/N) ratios.

The instrument was calibrated for ionization efficiency (IE), inlet flow rate, and particle
sizes using the standard procedure described by (Jayne et al., 2000). These three
calibrations were performed at the beginning, in the middle and end of the field study.
Particle-free ambient air was sampled at the end of the study to determine the detection
limits (DLs) of individual species and also for adjusting the fragmentation table. Default
relative ionization efficiency (RIE) values were assumed for organics (1.4), nitrate (1.1),
sulphate (1.2), and chloride (1.3), while an RIE value of 3.9 was determined for
ammonium following the analysis of pure $NH_4NO_3$. The close concentrations between
measured ammonium and predicted ammonium based on the stoichiometric charge
balance between nitrate, sulphate, and chloride (slope = 0.94, Fig. S4) suggest that these
RIE values are suitable for this campaign.

2.2.2  Operations of other instruments
The SMPS consisted of a condensation particle counter (CPC) (TSI, model 3772) and a
differential mobility analyser (DMA) was deployed at 5 min interval. Sample and sheath
flow rates of the DMA were set to 1 L min$^{-1}$ and 5 L min$^{-1}$, respectively. The SMPS was
calibrated using a polystyrene latex (PSL) standard prior to field measurements.

The SP2 uses an intra-cavity Nd:YAG laser at 1064 nm to determine the optical size of a
single particle by light scattering and, if material within the particle absorbs at this laser
wavelength, the refractory mass of the particle quantified by detection of the main light-



absorbing component is rBC. The SP2 incandescence signal was used to obtain single
particle rBC mass after calibration with Aquadag standard BC particles. The measured
rBC mass is converted to a mass equivalent diameter, which is termed as the BC core
diameter ($D_c$) - the diameter of a sphere containing the same mass of rBC as measured in
the particle. Any measured particle with a detectable incandescence signal is referred to
as an rBC particle, whereas a particle which only exhibits a scattering signal is termed as
a non-BC particle. The total rBC mass loading is reported as the sum of all detected
single particle rBC masses.

The aethalometer measures the optical attenuation (absorbance) of light from LED lamps
emitting at seven wavelengths (370, 470, 520, 590, 660, 880, and 950 nm) with a typical
half-width of 20 nm. The difference in light transmission through the particle-laden
sample spot and a particle free reference spot of the filter is attributed to the absorption
caused by aerosol. The attenuation of light is converted to the BC mass concentration
using wavelength-dependent calibration factors as recommended by the manufacturer.
BC was measured using data at 880 nm using a specific attenuation cross section of 16.6
$m^2$ $g^{-1}$ and an empirical correction factor (2.14) for the shadowing effect during the
campaign. The flow rate was maintained at 4.8 L $min^{-1}$ calibrated using a flow meter.
Detection limit of the aethalometer BC was determined to be 0.16–0.28 μg $m^{-3}$ with a
flow rate of 4.8 LPM and 5 min time interval, calculated as three times the standard
deviation (3σ) of the dynamic blanks. The TEOM was operated at a temperature of 40 °C
in order to minimize mass loss due to volatilization of semi-volatile aerosol compounds.
The time resolution of $PM_{2.5}$ mass concentration was 5 min.

2.3 Data processing
2.3.1 General AMS data processing
The HR-ToF-AMS data were processed using the standard software of SQUIRREL
(v1.56) and PIKA (v1.15c) (http://cires.colorado.edu/jimenez-
group/ToFAMSResources/ToFSoftware/index.html) to determine the mass



concentrations and the size distributions of the NR-PM$_1$ species and the ion-speciated
mass spectra of organics, written in IGOR (Wavemetrics, Inc., Lake Oswego, OR, USA).
An empirical particle collection efficiency (CE) of 0.5 was used, which has been widely
used in field studies employing AMS with a dryer installed in front of the equipment's
particle inlet. This CE value was further validated by the consistency between HR-ToF-
AMS and SMPS data ($R^2$ = 0.9, Slope = 1.48, Fig. 3). The elemental ratios of OA (O:C,
H:C, and OM:OC) for this study was determined using the "Aiken ambient" method
(Aiken et al., 2008) other than the "improved-ambient" method (Canagaratna et al., 2015)
which increased O:C on average by 27%, H:C on average by 10%, and OM:OC on
average by 7% (Fig. S2). These "Aiken ambient" results of elemental ratios are more
suitable here to allow for comparison with those during summer 2012.

2.3.2  Positive Matrix Factorization (PMF) analyses
The source decomposition of organics was analysed by PMF with the multilinear engine
(ME-2) algorithm which serves to reduce rotational ambiguity within the PMF2
algorithm. The ME-2 algorithm allows the user to add a priori information into the model
(e.g., source profiles) to constrain the matrix rotation and separate the mixed solution or
the weak solution. The PMF analysis of organic matrix using ME-2 algorithm is
implemented within the toolkit SoFi (Source Finder) and perform by the so-called a-value
approach (Canonaco et al., 2013). First, organic matrix was analysed using the PMF2.exe
algorithm in robust mode (Paatero and Tapper, 1994) and explored using the PMF
Evaluation Toolkit (PET) (Ulbrich et al., 2009). The PMF solution was evaluated
following the procedures outlined in Table 1 of Zhang et al. (2011a) including
modification of the error matrix and downweight of low S/N ions. Moreover, based on
the AMS fragmentation table, some organic ions were not directly measured but scaled to
the organic signal at $m/z$ 44, which were downweighted by increasing their errors by a
factor of 3. Some highly polluted periods were deleted during PMF analysis such as
January 22-23, 2014. The results of four, five, and six factor solutions with ƒPeak at 0 are
shown in supplementary material (Fig. S5-S7). It is easy to find that a hydrocarbon-like



OA (HOA) factor, a cooking-emitted OA (COA) factor, a semi-volatile and low-volatility
oxygenated OA (SV-OOA and LV-OOA) factors could be clearly separated in the four-
factor solution; for the HOA factor, there were significant contributions from $m/z$ 60, 73,
92, and 115 in the mass spectrum, suggesting a mixing of multiple sources. In the five-
factor solution, a coal combustion-emitted OA (CCOA) factor was separated; however,
$m/z$ 60 and 73 which are related to biomass burning OA (BBOA) could not be separated.
We then performed OA source apportionment using the ME-2 algorithm by constraining
the profiles of HOA and BBOA with the fixed a-value of 0.1 for HOA and 0.4 for BBOA.
The a-value test was performed following the technical guidelines presented in Crippa et
al. (2014). The reference profile of HOA was adopted from the HOA of the summer
study and the reference profile of BBOA was adopted from the nine-factor PMF solution
of this study.

The size distributions of individual OA factors were determined via a multivariate linear
regression technique (Ge et al., 2012). This algorithm assumes that each OA mass
spectrum is the linear superposition of the mass spectra of individual OA factors, whose
mass profiles are constant across the whole size range. Further details about the algorithm
can be found in Xu et al. (2014).

2.3.3    Radiocarbon ($^{14}$C) data analysis
In order to identify the origins of SOA, we conducted $^{14}$C analysis on four filter samples.
These filter samples were collected at the CAEERI site which is about 500 m away from
the LZU site (Fig. S1a). Filter samples were collected using a low volume $PM_{2.5}$ sampler
(16.7 L min$^{-1}$) during January 2014 with a 24 h sampling time in every week for each
filter (January 3rd, 8th, 15th, and 23rd, respectively) on pre-baked quartz filters. One
field blank filter was collected and analysed to correct the filter sample measurements.
Organic carbon (OC) was separated from the filters by combustion at 375 ℃ during 200
s in pure oxygen in a thermo-optical OC/EC analyser (Model 4L, Sunset Laboratory Inc,
USA) (Zhang et al., 2012). The carbon isotopic analysis was conducted by online


coupling of the OC/EC analyser with the accelerator mass spectrometry system
MICADAS at the University of Bern, Switzerland (Zotter et al., 2014; Agrios et al.,
2015). Fossil $^{14}$C measurement results were transferred into the non-fossil fraction ($f_{NF}$)
of OC using a conversion factor of 1.03.

For the apportionment of AMS-PMF OA factors using $^{14}$C data (Zotter et al., 2014), we
assume that all OC sources are represented by the six PMF factors and the $f_{NF}$ in NR-PM$_1$
was the same as that in PM$_{2.5}$. The OA mass of each PMF factor and total OA were first
converted to OC mass using the OM:OC ratios derived from its MS (OM:OC$_{HOA}$ = 1.29,
OM:OC$_{BBOA}$ = 1.5, OM:OC$_{COA}$ = 1.27, OM:OC$_{CCOA}$ = 1.37, OM:OC$_{SV-OOA}$ = 1.55,
OM:OC$_{LV-OOA}$ = 2.01, OM:OC$_{total}$ = 1.51). For the OC mass concentration of the AMS
factors, the following notations, hydrocarbon-like organic carbon (HOC), biomass
burning organic carbon (BBOC), cooking organic carbon (COC), coal combustion
organic carbon (CCOC), oxygenated organic carbon (OOC), total organic carbon from
AMS (TOC$_{AMS}$), were adopted in the following sections. An $f_{NF}$ value was assumed a
priori for the primary PMF factors HOC, BBOC, COC, and CCOC. The average $f_{NF}$ of
OOC is then derived by the equation below:

$f_{NF\_OOC} = (TOC_{NF\_AMS} - f_{NF\_HOC} \times HOC - f_{NF\_BBOC} \times BBOC - f_{NF\_COC} \times COC - f_{NF\_CCOC}$
$\times CCOC) / (SV\text{-}OOC + LV\text{-}OOC)$
Here HOC is assumed to originate from gasoline and diesel exhaust and contains
exclusively of fossil carbon, i.e., $f_{NF\_HOC}$ = 0; BBOC is estimated partly from fossil
carbon such as soft coal due to the popular usage in local residents, i.e., $f_{NF\_BBOC}$ = 1;
COC is assumed to originate from non-fossil carbon such as cooking oil and dressing, i.e.,
$f_{NF\_COC}$ = 1; CCOC is estimated to originat from coal combustion, i.e., $f_{NF\_CCOC}$ = 0.



**3    Results and discussions**
3.1    Overview of field study
3.1.1    Meteorological conditions
Fig. 1 shows the time series of meteorological parameters and $PM_1$ components during
January 10–February 4, 2014. During the campaign, the measurement site mainly
received air masses from northern and northwestern associated with low wind speeds (on
average: $0.8 \pm 0.4$ m s$^{-1}$). The wind directions were associated with the typical
anticyclone mesoscale weather forced by Tibetan Plateau in Lanzhou during winter. The
mountains to the north and south of the city significantly reduced the wind speed. Air
temperature was typical in winter of Lanzhou ranging from $-10$ to 14 °C (average = $0.5 \pm$
5.0 °C), but was a little warmer after the Chinese New Year (January 31, 2014) (Fig. 1a).
No precipitation event occurred during the campaign, and RH was pretty low ranging
from 8.8 to 50.7% (avg. = $27.7 \pm 9.1\%$). The visibility ranged from 3.7 to 50 km (avg. =
$16.0 \pm 8.7$ km).

3.1.2    Inter-comparisons
The inter-comparisons between AMS *vs*. SMPS and TEOM are shown in Fig. S3.
Comparison between the mass concentration of $PM_1$ and the volume of particle measured
by SMPS is tightly correlated ($R^2 = 0.9$) with a slope of 1.48, which represents the
average density of bulk particles, assuming that the AMS and the SMPS measure a
similar particle population. This value is indeed very close to the estimated $PM_1$ density
(1.46) based on the measured particle composition for this study (using density of 1.2 g
m$^{-3}$ for organics, 1.72 g m$^{-3}$ for $NH_4NO_3$, 1.77 g m$^{-3}$ for $(NH_4)_2SO_4$, 1.52 g m$^{-3}$ for $NH_4Cl$
and 1.8 g m$^{-3}$ for BC) (Zhang et al., 2005; Bond and Bergstrom, 2006). The mass
concentration of $PM_1$ is also closely correlated ($R^2 = 0.71$) with TEOM $PM_{2.5}$
concentrations with a slope of 0.73. Similar contribution of $PM_1$ to $PM_{2.5}$ were also
observed in other cities in China during winter (Elser et al., 2016), such as Beijing (0.74
during 2011) (Sun et al., 2013b). Note that the actual mass ratio between $PM_1$ and $PM_{2.5}$





should be higher than these values since refractory materials such as crustal components
were not measured.

3.1.3   $PM_1$ composition, variation, and acidity
The average mass concentration of $PM_1$ (NR-$PM_1$ + BC) was 57.3 μg m$^{-3}$ (ranging from
2.1 to 229.7 μg m$^{-3}$ for hourly average) during this study, with 51.2% of organics, 16.5%
of nitrate, 12.5% of sulphate, 10.3% of ammonium, 6.4% of BC, and 3.0% of chloride
(Fig. 2a). The average mass concentration was more than twice the average value
observed during summer 2012 (24.5 μg m$^{-3}$). All species showed similar day-to-day
variation with nitrate being the most significant one (Fig. 1d), suggesting an important
local source for nitrate. The mass contributions of $PM_1$ species from low to high $PM_1$
concentrations showed an increased contribution for organics (49% to 53%) and nitrate
(13% to 18%), but a decreased contribution for sulphate (17% to 11%) and BC (7.3% to
5.3%) suggesting somewhat different chemical processes/sources for each species during
the haze pollution (Fig. 2b). Specifically, the increased organics was mainly due to the
contribution of primary OA (POA) based on PMF analysis (more discussion are given in
section 3.5). During the late part of Chinese New Year holiday (February 3 to end of the
study), $PM_1$ concentration decreased in association with increased wind speed. NR-$PM_1$
appeared to be neutralized throughout this study, as indicated by an overall stoichiometric
charge balance between the anions (i.e., nitrate, sulphate, and chloride) and the cation
ammonium (slope = 0.94, Fig. S4). This result indicates that the inorganic particulate
species were mainly present in the forms of $NH_4NO_3$, $(NH_4)_2SO_4$, and $NH_4Cl$ in $PM_1$.

3.1.4   Size distribution
The average chemically-resolved size distributions of NR-$PM_1$ species are shown in Fig.
3a. While all components peaked between 400–500 nm, organic aerosol presented a
wider distribution than the inorganics and extended to ~250 nm, suggesting the influence
of fresh organics (primary OA, more discussion are given in section 3.4). These features
were similar to those found in most urban sites by the AMS. The similar mode size of





inorganics and SOA (Fig. 3c) suggested the well external mixed air mass during the
sampling period. The mass contributions of chemicals at the major peak (400–500 nm)
were organics (~50%), nitrate (~20%), ammonium (~15%), sulphate (~10%), and
chloride (~5%); while the contribution of organics increased with the decreasing of size
mode (Fig. 3c). Comparing with the results observed during 2012 summer, the size
distributions of aerosol particle during winter were narrower, although the mode sizes of
major peaks were similar, indicating highly mixed and aged aerosol particles during
winter.

3.2    Diurnal variations of aerosol species
All species show significant diurnal variations during the study suggesting the important
local and regional sources of aerosol (Fig. 4). The observed diurnal trends of BC
presented two dominant peaks with one at late morning (10:00–12:00) and another at
early evening (20:00–22:00). The morning peak did not overlap with the rush hours
(7:00–9:00), different than that of summer 2012; the BC mass loading started to increase
from 6:00 continuously during morning, and reached maximum between 10:00–12:00
and then dropped down after the noon time. Another combustion tracer, carbon monoxide
(CO), also showed the similar morning peak (Fig. 5). This morning peak was likely
resulted from the formation of inversion layer during winter at Lanzhou which promoted
accumulation of air pollutants from enhanced human activities in the morning. This
inversion layer frequently formed from night time and diffused after the noon time due to
the valley terrain. The temperature profile observed at the suburban Lanzhou (Yuzhong,
~30 km from the sampling site) showed a strong inversion in the low boundary layer
during the morning time (Fig. S8). The evening peak of BC could result from increased
human activities such as traffic, cooking, and heating coupled with low boundary layer
after sunset. Organics had two sharp peaks at the noon time (12:00–13:00) and early
evening (19:00–20:00) which correspond to lunch time and dinner time, respectively,
indicating the importance of cooking-related emissions of OA. PMF analysis show that





cooking-emitted aerosol could contribute up to 30 – 60% of organics during meal times
(section 3.4.3).

Sulphate presented two peaks with one occurring at the noon time (11:00–14:00) in
accordance with the photochemical processes; this peak is narrower than that during
summer, likely due to relatively weak photochemical activities. Another minor peak
occurred between 20:00–22:00 which was likely due to the lowered boundary layer depth.
The significantly higher concentration of sulphate during winter than summer suggests a
higher precursor $SO_2$ emission and stagnant atmospheric conditions in winter. The
diurnal pattern of sulphate during winter was similar to that of summer 2012 at Lanzhou
and summer 2011 at Beijing, but was different from that of Beijing during winter
2011/2012 where aqueous processing was found to could play an important role (Sun et
al., 2013b). Chloride had similar diurnal pattern with sulphate, although the evening peak
was more obvious. The major source of hydrochloric acid is biomass burning, coal
combustion and waste combustion (Ianniello et al., 2011). The significant evening peak
could be related with these sources coupled with the shallow boundary layer. The high
background concentrations of chloride during day and night suggest a persistent emission
of hydrochloric acid which could be from the heating factory and power plants. The
diurnal pattern of chloride during winter was different from that during summer 2012
which peaked during the night time due to temperature-dependent gas-particle
partitioning. Nitrate peaked between 12:00–16:00, right after the peak of sulphate. The
formation of nitrate during afternoon suggests that nitrate was dominated by the
homogeneous photochemical production. Fig. 5 shows the variations of $NO_x$ and $O_3$
calculated from data downloaded from one station monitored by the Ministry of
Environmental Protection of China, ~3 km southwest of sampling site (Fig. S1a); NO had
a morning peak (7:00–10:00) and an evening peak (19:00–21:00) corresponding to rush
hours; $NO_2$ increased from 10:00 which formed from NO consumed by OH radical and
decreased from 14:00 corresponding to the formation of nitrate and $O_3$ during afternoon.
The diurnal change of $NO_x$ ($\Delta NO_x$) mixing ratio was ~50 ppbv (from 150 to 100 ppbv),
while the diurnal change of the sum of $\Delta O_3$ and $\Delta NO_3^-$ was ~30 ppbv. Considering the



higher mixing layer height during afternoon, it seems that nitrate was mainly formed
from the photochemical processing of NOx. The diurnal pattern of nitrate during winter
was vastly different from that during 2012 summer which was mainly controlled by the
dynamic of mixing layer and gas-particle partitioning. It seems that atmospheric
ammonia was first neutralized by sulphuric acid and hydrochloric acid to form
ammonium sulphate and ammonium chloride, and the remaining ammonia may then
combine with nitric acid to form ammonium nitrate.

3.3    Bulk characteristics and elemental ratios of OA
Table 1 shows the average elemental mass composition and mass contributions of six ion
categories to the total organics. Carbon contributed 66% to the organics following by
oxygen (25%), hydrogen (8%), and nitrogen (1%); correspondingly, $C_xH_y^+$ dominated the
organics by 59%, following by $C_xH_yO_1^+$ (26%), $C_xH_yO_2^+$ (10%), $H_yO_1^+$ (2%), and
$C_xH_yN_p^+$ (2%). Compared with the results of 2012 summer, the organics in winter had
higher carbon (66% *vs*. 59%) and $C_xH_y^+$ content (59% *vs*. 56%), and lower oxygen
content (25% *vs*. 26%) (Fig. 6c); this suggests that the organics during winter had a
higher fraction of primary compounds than those during summer which is likely due to
weaker photochemical activities, lower boundary layer height and more emissions from
primary sources. The average O/C of organics, an indicator for oxidation state, was 0.28
during this study which was somewhat lower than that of summer 2012 (0.33) (Fig. 6a
and b). Photochemical processing of organics during winter appeared to be significantly
weaker and shorter than those during summer as shown by the diurnal pattern of O/C (Fig.
6d). The diurnal profile of H/C was inversely correlated with that of O/C, and the peaking
of organic aerosol concentration usually corresponded to the high H/C ratio and low O/C
ratio, indicating the dominant role of primary OA.

3.4    Source apportionment of OA
Source apportionment via PMF with ME2 engine on OA mass spectra resolved six
components, i.e., HOA, COA, CCOA, BBOA, SV-OOA, and LV-OOA. Each component



has a unique mass spectral pattern, diurnal pattern, and temporary variation which
correlated with corresponding tracers such as inorganic species. Two OOA components
can be regarded as surrogates of secondary OA (SOA), with LV-OOA for more aged
SOA and SV-OOA for fresher SOA; The HOA, BBOA, COA and CCOA components
are regarded as primary OA (POA) based on their low O/C ratios and good correlations
with primary aerosol tracers (Fig. 7). Comparison with the source apportionment results
of summer 2012, the organic sources and chemical processes during winter 2013/2014
were more complex due to the multiple primary sources. Detailed discussion of each
factor is given in the following subsections.

### 3.4.1  HOA
HOA factors had been frequently separated from the OA in urban area due to the
emission from traffic and/or other fossil combustion activities (e.g., Sun et al., 2011b; Ge
et al., 2012). The diurnal pattern of HOA in winter 2013/2014 of Lanzhou shows two
predominant peaks in the morning (10:00–12:00) and evening (20:00–21:00),
respectively (Fig. 5). The morning peak started from 6:00 was mainly associated with the
morning traffic rush hours, and it maximized at late morning associating with the
inversion layer as discussed in Section 3.2. The evening peak was relevant with the
evening rush hours and low PBL depth after sunset. The relatively low concentration
during afternoon was probably due to the high PBL depth as shown by the mass
concentration variations of BC. The correlation between HOA and BC was high ($r = 0.84$,
Fig. 7f and Table 2), as a big fraction of BC has been thought to emit from traffic
activities and commonly used as a tracer of traffic emission. The minimum of HOA
concentration, which typically occurred during afternoon or middle night, was still up to
~2 $\mu g\ m^{-3}$ suggesting a high background of HOA which is likely due to the stagnant air
condition unfavourable for the diffusion of aerosol. The size distribution of HOA showed
a mode size of ~200 nm (Fig. 3b) corresponding to the primary emitted aerosol
behaviours and HOA could account for ~25% mass of aerosols between 100-300nm (Fig.
3c). The average concentration of HOA during 2013/2014 winter was 2.6 $\mu g\ m^{-3}$



accounting for 9% of organics (Fig. 8a). This concentration was higher than that of 2012
summer in Lanzhou (2.6 *vs*. 1.8 µg m$^{-3}$) likely due to the lower PBL during winter and
stagnant air conditions. The mass contribution from HOA is similar to the result of 2013
winter at Beijing (9%) which was also the lowest contributor to the total OA (Sun et al.,
2013b; Zhang et al., 2014), probably due to more modern vehicles were used in the past
years.

### 3.4.2 BBOA
BBOA component had been widely observed in USA and European countries during
winter due to the traditional wood burning for residential heating (Alfarra et al., 2007).
The BBOA component is thought to be less important in China because coal is the major
fuel during winter. BBOA could be an important component in China during some
special periods. For example, Zhang et al. (2015a) identified a BBOA factor in urban
Nanjing, southeast of China, during harvest seasons of summer and autumn because of
the burning of straw. The BBOA component has also been identified in some regions in
China where the coal resource is scarce. For example, Du et al. (2015) separated a BBOA
factor at a rural site of the northern Tibetan Plateau due to the widely usage of cow dung
cake for heating in this region. The BBOA component has also been identified during
winter in cities in southern China because of rich wood resource in these regions (He et
al., 2011; Huang et al., 2011; Huang et al., 2013). To our knowledge, only two recently
papers have reported the identification of a BBOA factor during winter using online
measurement in an urban area of northern China (Elser et al., 2016). Although the high
contribution of non-fossil carbonaceous aerosol was found (Zhang et al., 2015) and the
mass spectra of organic in other cities (such as Beijing) during winter have also
significant contributions from $m/z$ 60 and 73 (Sun et al., 2013b; Zhang et al., 2014), it is
difficult to separate the BBOA using general PMF because of its similar temporal
variation with CCOA, such as diurnal pattern (Fig. 4). BBOA contributions presented a
clear periodic change (Fig. 1), and on average were high during night time and low
during daytime (Fig. 5). This trend is consistent with conventional usage of biomass for



heating. The time series of BBOA was also closely correlated with BC and chloride
(Table 2) due to significant emission of these species from biomass burning. The average
mass concentration of BBOA was 3.2 µg m$^{-3}$, on average contributing 11% of the total
OA mass for the entire study (Fig. 8a), but could reach up to 20% during night and down
to less than 5% during afternoon (Fig. 8b). This average concentration was close to the
results observed at southern Chinese cities such as Jiaxing (~3.9 µg m$^{-3}$) (Huang et al.,
2013), Kaiping (~1.36 µg m$^{-3}$) (Huang et al., 2011) and Shenzhen (~5.2 µg m$^{-3}$) (He et
al., 2011).

The size distribution of BBOA peaked at ~400nm which is close to accumulation mode
(Fig. 3b). This feature suggests that the BBOA factor observed during this study was
likely aged. This behaviour has been observed in some studies and two BBOA factors
were identified as a result: one fresh BBOA and one aged BBOA (Zhang et al., 2015a).
The relative high O/C and the dominance of an accumulation mode in the size
distribution of BBOA were also observed during winter in Fresno, a major city in the
Central Valley of California, USA (Ge et al., 2012; Young et al., 2015). These
observations are consistent with recent observations that levogolucosan, a major product
of biomass burning, can be quickly (within a few hours) oxidized once in the atmosphere
(Bougiatioti et al., 2014). Due to the mean RH during the whole study was lower than
30%, it seems that biomass emissions were oxidized mostly in the gas phase and the
oxidize products subsequently partitioned into the particle phase (Qin and Prather, 2006).

### 3.4.3  COA
The COA component has been widely identified in urban AMS studies and observational
results by other instruments recently, and it is regarded as important source of OA in
urban areas (Abdullahi et al., 2013 and references therein). The MS of COA in this study
had a major contribution from $C_xH_y^+$ ions (79.7%) with also an important contribution
from $C_xH_yO_1^+$ ions (15.5%), similar as those in HOA (80.6% and 13.7%) (Fig. S9). In
comparison with the HOA spectrum, COA had a higher $m/z$ 55 to 57 ratio (2.1 vs. 0.7)





(Fig. 7) which had been postulated as a significant indicator for COA (Sun et al., 2011b;
Mohr et al., 2012). In the V-shape plot defined by Mohr et al. (2012), which uses $f55$ *vs.*
$f57$ after subtracting the contributions from factors of OOA, CCOA, and BBOA (denoted
as OOA_CCOA_BOA_sub, i.e. $f55_{OOA\_CCOA\_BBOA\_sub}$ and $f57_{OOA\_CCOA\_BBOA\_sub}$), the data
can be clearly represented with ones during morning close to HOA line and ones during
meal times close to COA line (Fig. S10). The MS of COA is highly similar to that of
summer 2012 observation ($R^2$ = 0.95, slope = 0.97, Fig. S11) which was found to
resemble closely the COA MS from other locations (Xu et al., 2014). In fact, the COA
components were found to be associated with heating of cooking oils rather than burning
of meat/food itself, and indeed the COA mass spectra from cooking of different dishes
were highly similar (He et al., 2010). The O/C and H/C ratios of COA were 0.09 and 1.71,
respectively, suggesting its feature as POA. This O/C ratio was slightly lower (0.09 vs.
0.11) and the H/C was slightly higher than that of 2012 summer (1.71 vs. 1.69). The size
distribution of COA was also peaking between 100–200 nm similar to that of HOA (Fig.
3b). The diurnal variation of COA displayed two predominant peaks standing out at lunch
time (12:00–13:00) and dinner time (19:00–20:00), respectively (Fig. 5), and a small
breakfast peak (~8:00). This pattern was consistent with that of summer 2012 (Fig. 4)
which resulted from the consistent routine life during winter and summer. The enhanced
COA concentration at dinner time might be mainly due to the low PBL height and the
activity of a formal meal with more attendants and longer time than that of lunch. The
temporal variation correlated tightly with $C_6H_{10}O^+$ ($R^2$ = 0.95, Fig. 8d) which has been
reported as the high resolution mass spectral markers for ambient COA (Sun et al., 2011b;
Ge et al., 2012).

The average contribution of COA to organics was 20% (~10–50%) (Fig. 8a) with an
average mass concentration of 5.86 μg m$^{-3}$ which was much higher than those of HOA
and BBOA. This contribution is similar to those in Beijing during winter (average 19% of
OA with a range of 16–30%) (Sun et al., 2013b), Fresno (~19% of OA) (Ge et al., 2012),
Barcelona (17% of OA) (Mohr et al., 2012), and Paris (11–17%) (Crippa et al., 2013).



This high fraction indicates that COA is an important local source of OA in Lanzhou
regardless of clear or hazy periods (section 3.5).

3.4.4   CCOA
A CCOA component had been identified in this study with its MS similar to the OA from
coal burning in lab study (Dall'Osto et al., 2013). The MS of CCOA had high signals at
$m/z$ 41, 43, 44, 55, 57, 69, 91 and 115 (dominated by $C_xH_y^+$ ions) (Fig. 7i) (Elser et al.,
2016). $C_xH_y^+$ ions in total account for 75.3% of CCOA MS, following by $C_xH_yO_1^+$
(14.7%) and $C_xH_yO_2^+$ (9.4%). The fractions of $C_xH_y^+$ and $C_xH_yO_1^+$ were similar with
those in HOA MS (Fig. S9), but the CCOA MS had high signal intensity at $m/z$ 44
(mainly $CO_2^+$) which is different from that of HOA (Fig. 7). This high $CO_2^+$ fraction was
also observed in CCOA MS in Changdao island in China during winter (Hu et al., 2013).
Wang et al. (2015) suggested this high $CO_2^+$ signal is from the oxidative transformation
of the pyrolysis products during coal burning. Zhang et al. (2008) reported that 48–68%
of particulate organic matter from coal combustion aerosol is found in the form of
organic acids. The O/C ratio is thus higher than that of HOA (0.17 $vs.$ 0.10) with a lower
H/C ratio (1.67 $vs.$ 1.73). The CCOA also locates in a relatively high and left position in
the triangle plot defined by Ng et al. (2010) (Fig. 13a). These features indicate CCOA is a
POA factor but is more oxygenated than HOA. The time-dependent concentrations of
CCOA correlated with BC ($r = 0.74$) and chloride ($r = 0.62$) which also correlated well
with HOA and BBOA (Table 2). The CCOA mass loading remained high from 20:00 to
10:00, slowly decreased to a minimum at 16:00, and then increased from 16:00 to 20:00
(Fig. 5). This diurnal pattern was similar to that of BBOA which were all mainly emitted
from heating. The slower decreasing rate during morning and increasing rate during late
afternoon for CCOA than those of BBOA could related with wide usage of coal, such as
cooking and power plants. In our summer 2012 observation, we also observed OA signals
from coal combustion which have been persistent emitted during the whole year in
Lanzhou. The size distribution of CCOA peaked ~450 nm (Fig. 3b), similar with that of
BBOA.




The average CCOA mass concentration was 5.3 μg m$^{-3}$, accounting for 18% of total OA
mass (Fig. 8a). The mass fraction of CCOA could reach to 25% of OA during night and
decreased to 3% during afternoon (Fig. 8b). This indicates that CCOA was an important
OA component similar as that in Beijing OA (15–55%) (Zhang et al., 2014; Elser et al.,
2016), but its mass fraction of $PM_{2.5}$ (~9%) was at the low end of the values observed at
Beijing and Xi'an (9–21%) (Huang et al., 2014).

3.4.5   SV-OOA and LV-OOA
Two or more OOA components are commonly separated by PMF in urban areas which
correspond to fresh SOA and aged SOA (Jemenez et al., 2009), and the MS of SOA
factors all have predominant contributions at $m/z$ 43 and 44. The MS of fresher SOA such
as SV-OOA has higher contribution at $m/z$ 43 (mainly $C_2H_3O^+$, accounting for 73% of
$m/z$ 43 in this study), while aged SOA such as LV-OOA has higher signal at $m/z$ 44
(mainly $CO_2^+$, accounting for 98% of $m/z$ 44 in this study). The contribution of $C_xH_yO_1^+$
in SV-OOA was 35.6% followed by $C_xH_y^+$ (50.1%), $C_xH_yO_2^+$ (9.4%), $H_yO_1^+$ (1.4%),
$C_xH_yN_p^+$ (3.0%), and $C_xH_yO_zN_p^+$ (0.4%) (Fig. S9). The O/C ratio of SV-OOA was 0.31
and H/C was 1.47 consistent with fresh SOA. The MS of LV-OOA was comprised of
22.1% of $C_xH_yO_2^+$, 35.6% of $C_xH_yO_1^+$, 34.5% of $C_xH_y^+$, 5.5% of $H_xO_1^+$, 2.0% of $C_xH_yN_p^+$,
and 0.4% of $C_xH_yO_zN_p^+$ (Fig. S9). The O/C and H/C ratios of LV-OOA were 0.67 and
1.29, respectively. Compared with those of summer 2012, both O/C and H/C ratios of
SV-OOA during winter 2013/2014 were higher (0.31 $vs$. 0.28 for O/C, 1.47 $vs$. 1.34 for
H/C), while they were both slightly lower for LV-OOA (0.67 $vs$. 0.68 for O/C and 1.29 $vs$.
1.34 for H/C). These results indicate that the atmospheric oxidation capacity during
winter was still very strong. The positions of SV-OOA and LV-OOA in triangle plot of
$fCO_2^+$ $vs$. $fC_2H_3O^+$ are situated in the upper left corner (Fig. 13a), respectively, suggesting
the oxidation evolution of OOA. The MS of SV-OOA and LV-OOA were similar with
those of summer 2012 ($R^2$ = 0.98 for LV-OOA and $R^2$ = 0.77 for SV-OOA, Fig. S11).
Note that the $C_xH_y^+$ ions in SV-OOA were mainly from by $m/z$ 39, 41, 91 and 115 (Fig.



7h), which were also found to be enriched in coal combustion organic aerosols. This
feature is similar to that of summer 2012, potentially suggesting that part of SV-OOA
was from further oxidation of CCOA.

The temporal variations of SV-OOA and LV-OOA were highly correlated with
secondary inorganic species: SV-OOA *vs*. sulphate ($R^2 = 0.79$) and LV-OOA *vs*. nitrate
($R^2 = 0.71$) (Fig. 7a and b, Table 2). These patterns are somewhat contradictory to
previous AMS findings that SV-OOA typically correlates better with nitrate due to their
similar semi-volatile characteristics while LV-OOA tends to correlate better with
sulphate as they are both low-volatility species. These correlations were indeed observed
during the summer study of 2012 (Xu et al., 2014). The behaviours of the two OOA
factors during this study were likely due to the low air temperature and low RH
conditions which favoured secondary aerosol formation primarily through gas reactions.
Ammonium sulphate was quickly formed in the atmosphere than nitrate, and when $SO_2$
was completely consumed, ammonium nitrate would be formed. The concentration of
$NO_x$ was higher than $SO_2$ (120 *vs*. 35 ppbv) which the consumption of oxidants by
sulphate was relatively less. This phenomenon was also observed in winter time of
Beijing which showed strong photochemical formation of nitrate during afternoon (Sun et
al., 2013b).

The diurnal variation profiles of SV-OOA and LV-OOA all showed one bump with the
SV-OOA peaking between 11:00–14:00 and LV-OOA peaking between 12:00–18:00,
suggesting the importance of photochemical processes for both OOA factors. The size
distribution of the OOA (SV-OOA + LV-OOA) had a mode size of ~550 nm (Fig. 3b)
reflecting the feature as SOA. This size mode is slightly bigger than those of OOA in
other studies such as Fresno (460 nm) and Lanzhou summer 2012 (~450 nm) likely due
to the high concentration of gas precursors and longer lifecycle of aerosol during winter.



The mass concentrations of SV-OOA and LV-OOA were 7.0 and 5.6 μg m$^{-3}$ with the
mass contributions of 24% and 19% to OA, respectively (Fig. 8a). These contributions
were lower than those during summer 2012 in Lanzhou (27% for SV-OOA and 32% for
LV-OOA) especially for LV-OOA, likely due to the relative weak solar radiation during
winter and more primary sources in winter. The diurnal total contribution of OOA (SV-
OOA + LV-OOA) varied between 40%−70% (Fig. 8b), suggesting the importance of
SOA in the air pollution throughout the day at Lanzhou.

3.5    Primary and secondary OA
As shown in Fig. 2b, the mass fraction of organics increased with the increase of PM$_1$
concentration, so it is important to know the relative contributions of primary and
secondary OA components during the pollution periods. Fig. 9a shows the scatter plot of
SOA (= SV-OOA + LV-OOA) and POA (= HOA + BBOA + COA + CCOA) during this
study. It is clear that POA and OOA show correlation during the periods of POA less
than ~15 μg m$^{-3}$ associated with low mass fractions of OA. When POA and OA fraction
increased significantly, POA and OOA show almost no correlation, indicating the
importance of POA in the severe aerosol pollutions in Lanzhou during winter. This is
different than the observation from summer 2012, during which SOA had a stable
contribution to PM$_1$ (Fig. 9b), due to more complex POA sources and larger contributions
from these sources to PM$_1$ mass loading during winter compared to summer. This is even
more evident when comparing each POA factor with OA (Fig. 10). The COA had the
biggest contribution to the increased organics can explained 56% of the increase of
organics, followed by HOA (28%). The components of CCOA and BBOA also had
positive contributions to the increase of PM$_1$ mass. However, both OOA components had
negative slopes with organics with SV-OOA being the major one, likely suggesting that
the formation of SV-OOA might be limited during heavy haze period in Lanzhou since
the reduction of solar radiation may to some extent weaken the photochemical oxidation
activities. The phenomenon of POA dominating during haze periods is different from the
results in other cities in China (Huang et al., 2014). For example, Elser et al. (2016)



found significant increased contribution from SOA and secondary inorganic aerosol
during haze periods in 2013/2014 winter in Xi'an and Beijing. This is likely due to the
higher RH values in the eastern China which is more favourable for the aqueous-phase
production of SOA. Indeed, Sun et al. (2013a) observed significant increase of secondary
inorganic aerosol during high RH periods in Beijing.

The average contribution of POA to organics decreased from 60.0% to 39.3% during
Chinese New Year festival of 2014 (Fig. 1) due to the reduced primary aerosol sources
such as HOA (9.8% to 3.3%), COA (21.1% to 11.6%), CCOA (18.2% to 15.4%), and
BBOA (10.8% to 9.0%). This is an indication that control of cooking activities and traffic
emissions in this residential area may be effective strategies for air quality improvement
during winter.

3.6   Fossil and non-fossil OC
OC measured by OC/EC analyser on two filters ($OC_{filter}$) and corresponding AMS
($OC_{AMS}$) online measured results are shown in Fig. 11a. The average ratio of
$OC_{AMS}/OC_{filter}$ was ~1.5 for these two filters likely due to the analytical uncertainties of
different instruments (30% for AMS and 20% for $OC_{filter}$), which was also observed in
other studies (Zotter et al., 2014). The data from the $^{14}C$ measurement for the filter
samples are listed in Table S1. The total average of $f_{NF}$ in these four filters was $55 \pm 3\%$,
with 54% and 57% for filters during Jan. 15 and Jan. 23, respectively. Comparison with
other studies, the average $f_{NF}$ value in this study was lower than those in Xi'an (63%) and
Guangzhou (65%), and higher than those in Beijing (42%), while similar with those in
Shanghai (51%) during 2012/2013 winter (Zhang et al., 2015). Combining with the $f_{NF}$
value (the total average of $f_{NF}$ for the total average AMS results) and the contributions of
fossil (F) POC (HOC and CCOC) and non-fossil (NF) POC (BBOC and COC), the $f_{F}$ and
$f_{NF}$ for SOC could be obtained (Fig. 11b). The average $f_{F}$ and $f_{NF}$ for POC and SOC are
summarized into Fig. 12. The $f_{F}$ and $f_{NF}$ for POC during Jan. 15 were 47% and 53%,
while for SOC were 44% and 56%. For all AMS data, the $f_{F}$ and $f_{NF}$ in POC were 47%



and 53%, while for SOC were 43% and 57%. The F-POC during Jan. 15 was comprised
by 21% HOC and 26% CCOC, and NF-POC by 16% BBOC and 37% COC. For all AMS
data, the F-POC was comprised by 16% HOC and 31% CCOC, and NF-POC by 17%
BBOC and 36% COC.

3.7    Evolution of OA and relationship between odd oxygen and SOA
The evolution of OA chemical composition upon aging has been an important subject
which is used to understand the formation of SOA. The methods to characterize this
evolution include the application of several specific diagrams, such as the AMS triangle
plot ($f$44 $vs.$ $f$43 or $f$CO$_2^+$ $vs.$ $f$C$_2$H$_3$O$^+$) (Ng et al., 2010). $f$44 is a tracer for aged OA
(mostly CO$_2^+$), while $f$43 (mostly C$_2$H$_3$O$^+$, with some contribution from C$_3$H$_7^+$) is mainly
associated with freshly formed SOA and POA. POA factors are usually located towards
the lower and lower-left corner in triangle plot, and with the aging, move up toward the
region of SV-OOA indicating by the increased $f$43 and $f$44; and with further aging, OA
move toward the region of LV-OOA indicating by the increased $f$44 and decreased $f$43. In
the plot of $f$44 $vs.$ $f$43 of this study (Fig. 13a), the data distributed in a narrow space and
move up vertically in the triangle space suggesting significant increasing in $f$44. The data
from the low (night time) to the high (afternoon time) $f$44 value corresponded to the
evolution of the photo radiation intensity suggesting the photochemical processes. The
LV-OOA lay at the top of the data consistent with its highly oxidized feature, while
BBOA, COA, and HOA all lay at the bottom of triangle space. CCOA is above those of
other primary factors consistent with the results of high acid contents in coal burning OA.
In the plot of $f$CO$_2^+$ $vs.$ $f$C$_2$H$_3$O$^+$ (Fig. 13b), most of data moved out of triangle space
because of the high contribution of C$_3$H$_7^+$ at $m/z$ 43, especially for data during night time.
$f$CO$_2^+$ and $f$C$_2$H$_3$O$^+$ both increased before the noon time, after that $f$C$_2$H$_3$O$^+$ stopped at
~0.05 and $f$CO$_2^+$ kept increase likely suggesting the evolution of SV-OOA to LV-OOA.
In comparison to the results in summer 2012, the data in winter were more concentrated
in the triangle space suggesting air masses with similar source contribution during winter.





In order to understand the possible sources of oxygenated OA, we also compared the
diurnal variations between LV-OOA and $O_x$ (Fig. 13c). Both $O_x$ and OOA are products of
photochemical reactions and the comparison between $O_x$ and OOA can offer insight into
the evolution of OA due to the dependence of the ratio on the VOC species (Herndon et
al., 2008), assuming aqueous processing and night time oxidation for OOA were less
important, such as during this study due to the low RH. High SOA *vs*. $O_x$ slopes were
observed (larger than 0.12 μg m$^{-3}$ ppb$^{-1}$) where aromatic VOC dominated the
photochemical processing, while a low slopes (~0.03 μg m$^{-3}$ ppb$^{-1}$) were observed where
alkene VOCs dominated the photochemical processing (Wood et al., 2010; Hayes et al.,
2013). Fig. 13c shows the scatter plot between $O_x$ and LV-OOA and sized by the mass
concentration of BBOA. $O_x$ and LV-OOA showed tight correlation ($R^2$ = 0.9) with a
slope of 0.18 μg m$^{-3}$ ppb$^{-1}$ which suggested the aromatic VOCs may be a large
contributor to SOA formation. This result was consistent with the distribution of VOC
components in China where aromatic VOC is dominant among VOCs in northern China
(Zhang et al., 2015b). Liu et al. (2012) suggested that aromatics emissions in Chinese
cities had been underestimated in models by a factor 4-10 which could lead to ~50%
increase of SOA production. Apart from the sources of coal combustion, biomass
burning, and traffic for aromatic VOC emission (Liu et al., 2008), cooking activities had
recently been concerned for accounting for aromatic VOC loading. He et al. (2015)show
that Chinese cooking can emit amount of alkanes (41.75%), alkenes (27.23%), and
aromatics (28.35%). Combining with $^{14}$C results, it suggested that sources that emit
modern carbon (e.g., cooking and biomass burning) seem to be higher emission of
aromatic VOC than traffic and coal combustion together.

## 824   4   Conclusions

In order to understand the sources and chemical processes of the air pollution during
winter in Lanzhou, a field study was conducted at an urban site of Lanzhou during
January 10 – February 4, 2014 using a suit of on-line instruments. The results show that
the average mass concentration of $PM_1$ (NR-$PM_1$ + BC) was 57.3 μg m$^{-3}$ (ranging from



2.1 to 229.7 μg m$^{-3}$ for hourly averages), with 51.2% of organics, 16.5% of nitrate,
12.5% of sulphate, 10.3% of ammonium, 6.4% of BC, and 3.0% of chloride. These mass
loading levels and chemical compositions were similar to those observed in Beijing
during winter. The mass concentration of nitrate and organics increased with the increase
of PM$_1$ loading, while sulphate decreased, indicating the importance of OA and nitrate
during severe air pollution. The size distributions of all the species displayed a moderate
size at 400–500 nm, suggesting that aerosol particles were largely externally mixed
during winter. All species presented significant diurnal variations. BC had two peaks at
10:00–12:00 and 20:00–22:00, respectively. Further analysis indicated that the first peak
was resulted from the inversion layer during morning which accumulated the air
pollutants from early morning and until the break-up at around noon time. The evening
peak of BC was related to human activities such as traffic and coal combustion coupled
with the shallow PBL. OA presented two peaks corresponding to lunch and dinner time
suggesting cooking to be an important source. Sulphate peaked during the noon time
(11:00–14:00) indicating the importance of photochemical processes. Nitrate presented
an afternoon peak (12:00–16:00) which indicate the photochemical processing of NOx.
PMF analysis of organic mass spectrum with the ME-2 engine identified six organic
aerosol sources: i.e., HOA, BBOA, COA, CCOA, SV-OOA, and LV-OOA. POA, which
includes HOA, BBOA, COA, and CCOA, accounted for 57% of OA mass and showed an
increased in concentration with the increase of PM$_1$ loading. This is an indication that
POA emission was one of the main reasons for the occurrence of heavy air pollution
episodes. The temporal profile of LV-OOA tightly correlated with that of nitrate, while
those of SV-OOA with sulphate correlated. This observation was different than those
observed during other studies and during summer at Lanzhou, indicating the importance
of photochemistry for nitrate during winter in Lanzhou due to cold air temperature and
low RH conditions. $^{14}$C analysis of OOC indicated that 57% of the SOC was formed from
non-fossil VOC source including biomass burning and cooking.



**Acknowledgements**
The authors thank their colleagues for continuing support and discussion. This research
was supported by grants from the Chinese Academy of Sciences Hundred Talents
Program, the Key Laboratory of Cryospheric Sciences Scientific Research Foundation
(SKLCS-ZZ-2015-01), the National Natural Science Foundation of China Science Fund
for Creative Research Groups (41121001, 21407079, 91544220), and the Chinese
Academy of Sciences Key Research Program (KJZD-EW-G03).



Atmospheric Chemistry and Physics Discussions — Open Access — EGU

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




Table 1 Comparison of the composition of category ions and elemental composition of
OA between winter 2013/2014 and summer 2012.

| Category Ions | Winter 2014 | Summer 2012 |
|---|---|---|
| $C_xH_y^+$ | 61% | 56% |
| $C_xH_yO_1^+$ | 25% | 27% |
| $C_xH_yO_2^+$ | 9% | 11% |
| $C_xH_yN_p^+$ | 3% | 3% |
| $C_xH_yN_pO_z^+$ | 0 | 1% |
| $H_yO_1^+$ | 2% | 2% |
| Elemental composition | | |
| C | 66% | 59% |
| H | 8% | 7% |
| O | 25% | 26% |
| N | 1% | 1% |


Table 2 Correlation coefficient ($r$) between time series of OA factors and other aerosol
species.

| $r$ | HOA | BBOA | COA | CCOA | SV-OOA | LV-OOA | POA* | SOA* |
|---|---|---|---|---|---|---|---|---|
| BC | **0.84** | 0.75 | 0.43 | **0.74** | 0.59 | 0.14 | **0.75** | 0.44 |
| PAH | 0.69 | 0.75 | 0.46 | 0.73 | 0.27 | -0.07 | **0.75** | 0.13 |
| Sulphate | 0.69 | 0.35 | 0.32 | 0.32 | **0.89** | 0.56 | 0.49 | **0.84** |
| Nitrate | 0.35 | 0.06 | 0.31 | 0.06 | **0.77** | **0.84** | 0.31 | **0.90** |
| Chloride | 0.75 | 0.62 | 0.33 | 0.62 | **0.79** | 0.30 | 0.61 | 0.64 |
| Sulphate + Nitrate | 0.50 | 0.17 | 0.34 | 0.16 | **0.86** | **0.79** | 0.40 | **0.93** |
| Sulphate + Nitrate + Chloride | 0.59 | 0.29 | 0.35 | 0.29 | **0.89** | **0.70** | 0.47 | **0.91** |

*  POA = HOA + BBOA + COA + CCOA, SOA = SV-OOA + LV-OOA





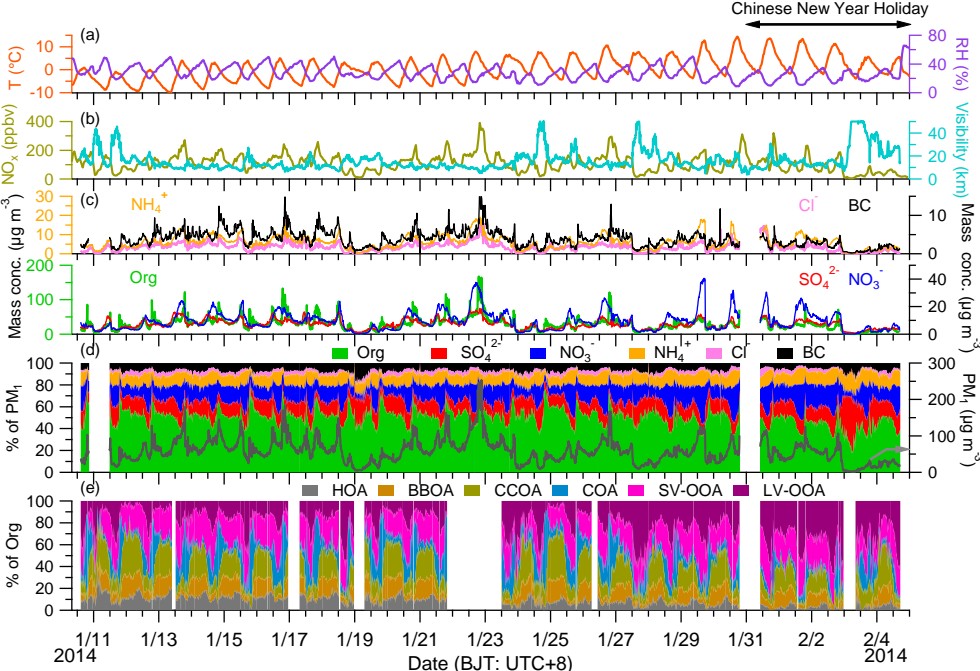

Fig. 1 Summary of meteorological and aerosol species data. (a) air temperature (T) and relative humidity (RH), (b) NOx and visibility, (c) mass concentration of PM$_1$ species, (d) the mass contribution of PM1 species, and (e) the mass contribution of organic components to organic aerosol.






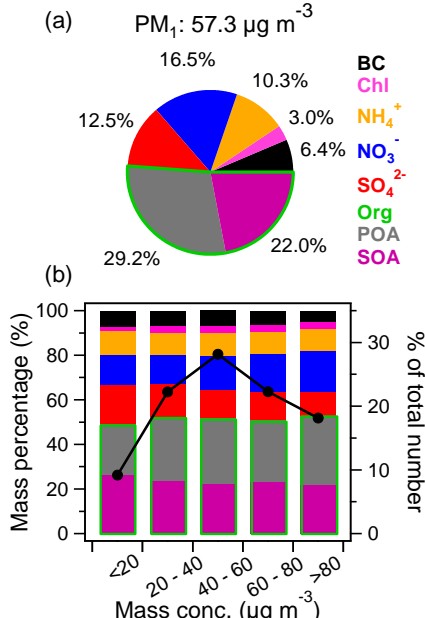

Fig. 2 The average mass contribution of PM$_1$ (= NR-PM$_1$ + BC) species (a) during the
whole sampling period and (b) as a function of the PM$_1$ mass concentration ($\mu$g m$^{-3}$) bins
(left). The right axis in (b) shows the accumulated data number in each bin. The organics
were decomposed into primary oganic aerosol (POA) and secondary organic aerosol
(SOA) using PMF (section 3.4).





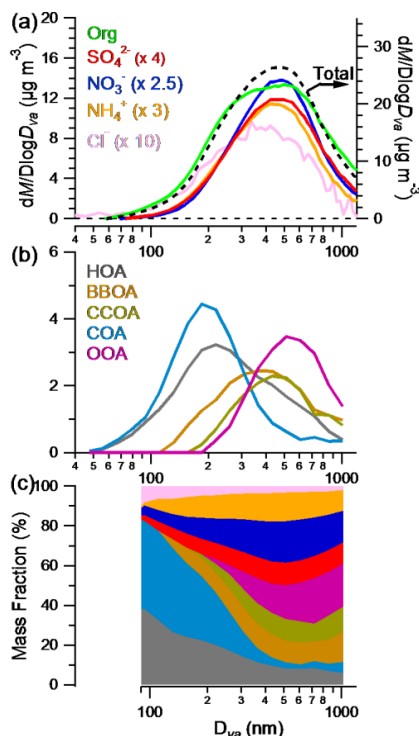


Fig. 3 The size distributions of (a) NR-PM$_1$ species, (b) organic components, and mass
contribution of all species to NR-PM$_1$.





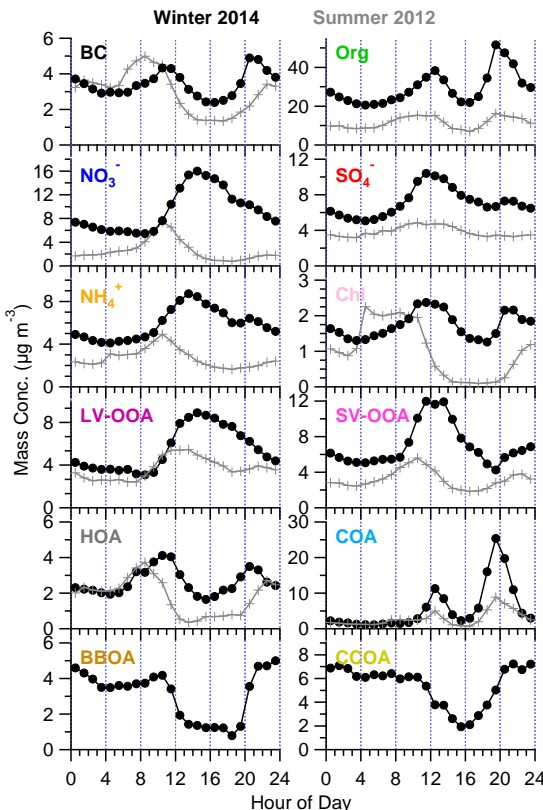


Fig. 4 The diurnal variation of PM$_1$ species during winter 2013/2014 and summer 2012.






Fig. 5 The diurnal variations of gas species downloaded from MAP-China station during
winter 2013/2014 and summer 2012.

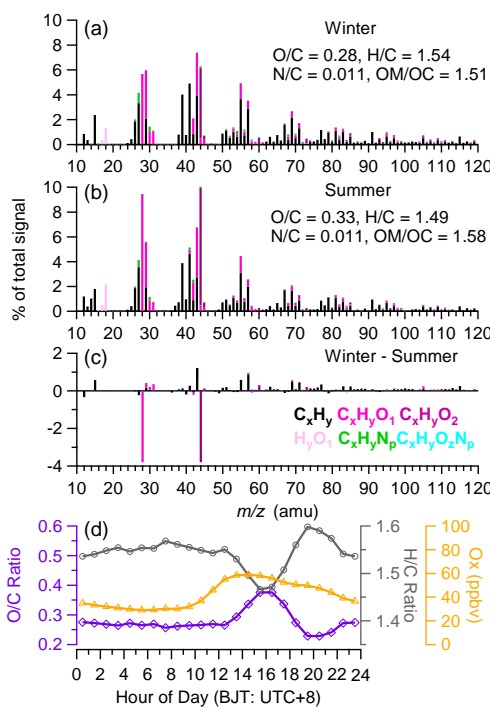


Fig. 6 The average HR-MS and elemental ratios of organics for (a) this study, (b) summer
2012, (c) the HR-MS difference between  this study and summer 2012, and (d) the
diurnal variations of elemental ratios and odd oxygen ($Ox = NO_2 + O_3$).





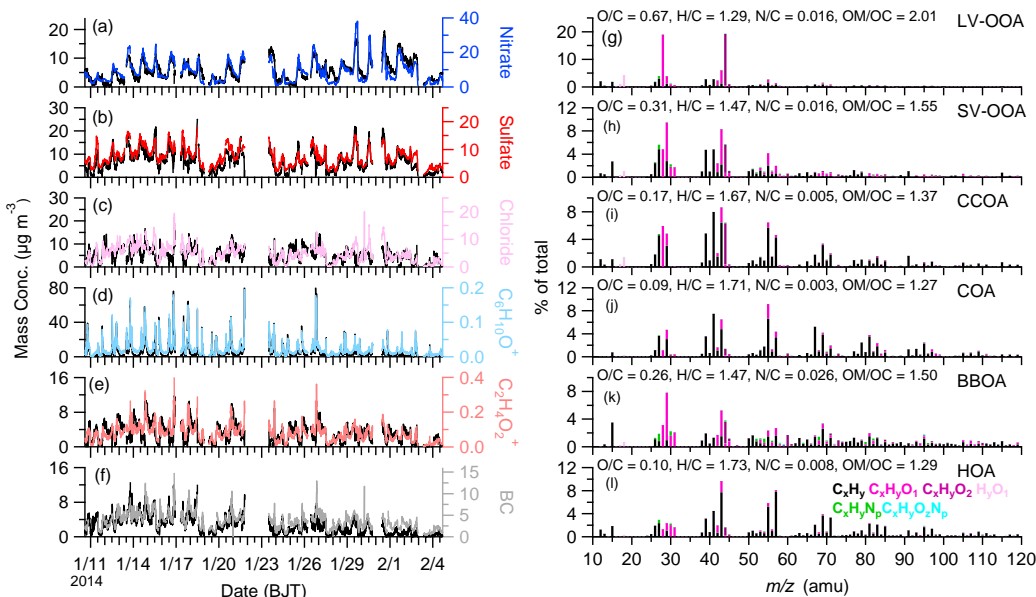


Fig. 7 The PMF results of time series (a – f) and HR-MS (g – l) for each component. The
temporal variations of different tracers are also present for supporting each component.

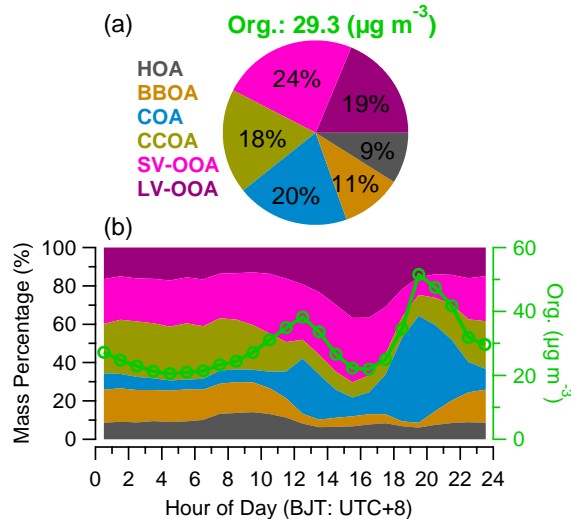


Fig. 8 (a) The average mass concentration of organics and mass contributions of organic
components to organics, and (b) the diurnal variations of organic components and
organics.





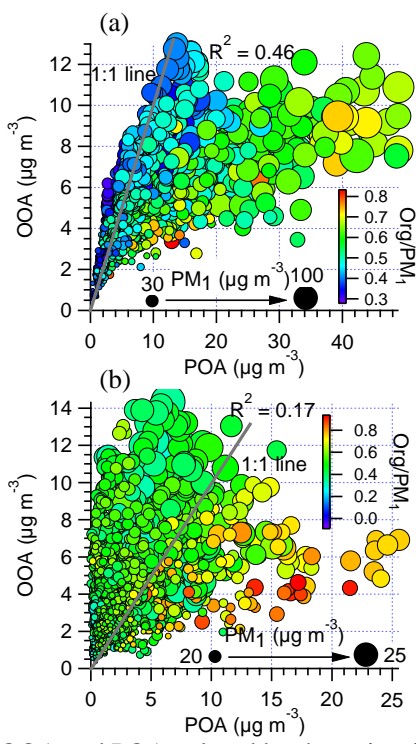

Fig. 9 The scatter plot of OOA and POA colored by the ratio of Org/PM$_1$ and sized by the
mass concentration of PM$_1$ for (a) winter 2013/2014 and (b) summer 2012.





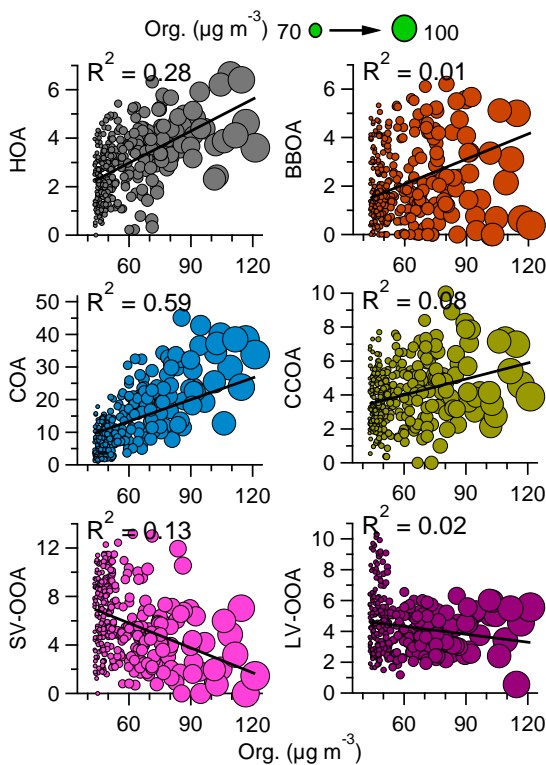


Fig. 10 The scatter plots of each organic component ($\mu g \ m^{-3}$) versus organics during haze
periods (definite as organics > 43 $\mu g \ m^{-3}$ (Org_avg + 1$\sigma$))





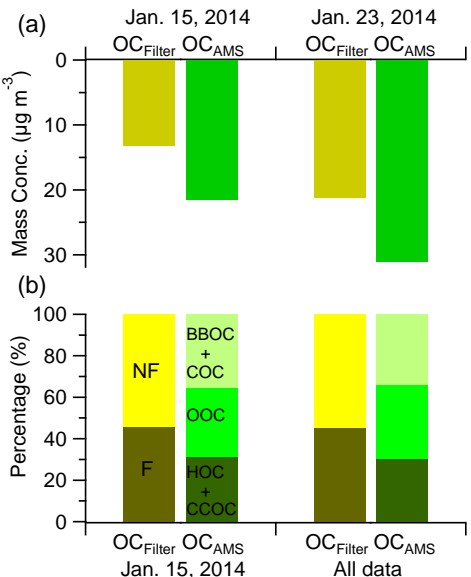

Fig. 11 (a) The comparisons of (a) OC concentration measured by filter sample (OC_{Filter})
and AMS (OC_{AMS}) on Jan. 15 and 23, 2014 and (b) the non-fossil (NF) and fossil (F)
carbon fraction measured by $^{14}$C and OC components in AMS.



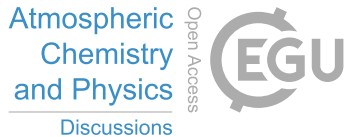

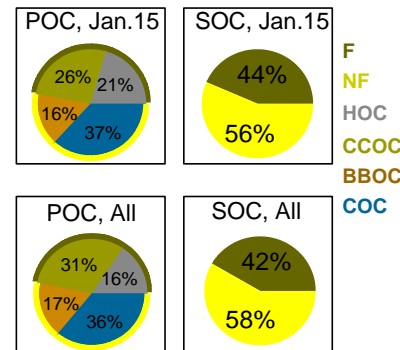


Fig. 12 The non-fossil (NF) and fossil (F) carbon fraction in POC and SOC during Jan.

1290                        15 and all data of AMS.




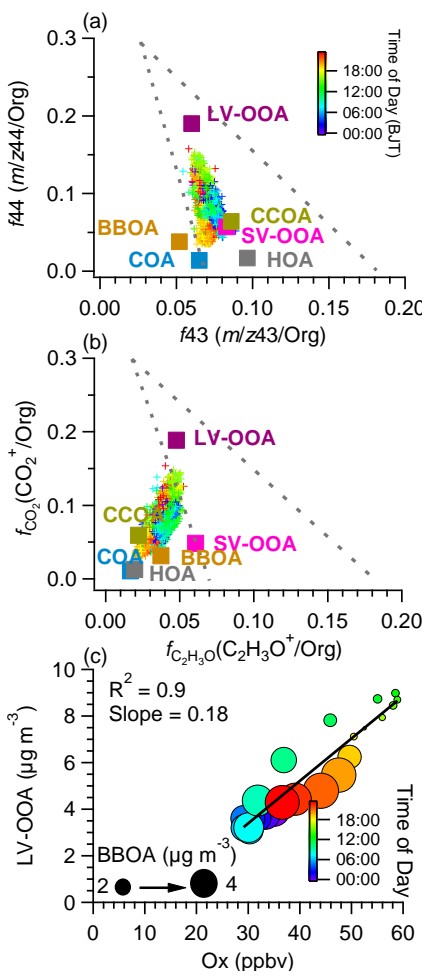


Fig. 13 Triangle plot of $f44$ (fraction of $m/z$ 44 in organics) vs $f43$ (fraction of $m/z$ 43 in
organics) and (b) $fCO_2^+$ (fraction of $CO_2^+$ in organics) *vs*. $fC_2H_3O^+$ (fraction of $C_2H_3O^+$ in
organics) for OA and six OA factors, and (c) scatter plot of LV-OOA *vs*. $O_x$ (the sum of
$O_3$ and $NO_2$) with linear fit and colored by time of day.
