# Peer review of "Wintertime organic and inorganic aerosols in Lanzhou, China"

_Atmospheric Chemistry and Physics, 2016_

## Referee Comment (RC1) · Anonymous Referee #1 · 8 Jun 2016

The manuscript by Xu et al. has a comprehensive characterization of wintertime sub-micron aerosols in Lanzhou, a highly polluted city in western China. The sources of organic aerosols were investigated using positive matrix factorization and radiocarbon analysis. The results were also compared with those measured in summer. This is an important study by providing scientific data and improving our knowledge on aerosol chemistry in western China. This manuscript fits within the scope of ACP, and I recommend it for publication after addressing the following minor comments.

Comments: 1. Please consider the usage of SV-OOA and LV-OOA since this study did not measure the OA volatilities. 2. As shown in Figure 2, POA showed an elevated contribution to PM at high mass loadings. Is this due to cooking OA, HOA or BBOA?

[Figure]

3. Line 41-43: please show the range for the changes in fractions of OA, nitrate and sulfate. 4. Line 46-47: frequently calm and stagnant air conditions during wintertime in Lanzhou. 5. Line 51: present the O/C value during summer 2012. 6. Line 60: haze to air pollution events. 7. Line 61-62: remove "The primary OA sources were more complex during winter than during summer". 8. Line 84-86: please add one or more citation. 9. Line 90-93: remove this sentence because of the duplication information with line 85. 10. Line 99: insert "recent" before "investigated..." 11. Line 119: add receptor model after (CMB) 12. Line 122-125: add the citation. 13. Line 142-146: combine these two sentences into one sentence. 14. Line 155: remove the comma after HR-ToF-AMS and add "of" after characteristics 15. Line 160: change basin to valley 16. Line 161-164: rephrase these two sentences. Two "thus" have been used which is a little bit confused. 17. Line 180: add "on average" before 0.82 18. Line 190: RH appears at the above content. 19. Line 200: instruments. 20. Line 246: consisted to consisting. 21. Line 270-271: where is this number (2.14) from, please add the citation. 22. Line 275: add the dry the aerosol in the sentence. 23. Line 310: show the reason to remove the period of Jan.22-23. 24. Line 315-317: how about seven solution results, could BBOA be separated? 25. Line 373-374: there is no wind data in Figure 1, please add them. 26. Line 412: please show the wind data. 27. Line 422: primary OA to POA. 28. Line 443-445: please add the citation. 29. Line 459: change suggest to could result from 30. Line 485-488: add "from the diurnal variation". 31. Line 498: is to was. 32. Line 502-504: Please add the diurnal variation of O/C in summer 2012. 33. Line 543: add "for all the PMF factor". 34. Line 544: change "in the past" to "in the recent". 35. Line 645: add "a little" before "more". 36. Line 653: change "have been" to "suggest".

---

## Referee Comment (RC2) · Anonymous Referee #2 · 1 Jul 2016

*Review of Xu et al. "Wintertime organic and inorganic aerosols in Lanzhou, China…"*

**Summary**

This manuscript summarizes measurements of aerosol composition, concentration, and size distributions carried out during wintertime in Lanzhou, China. A large majority of the data is derived from HR-ToF-AMS measurements, although data from an aethalometer, a SP2, a SMPS, and a TEOM are also reported in the manuscript. While the data are interesting and of high quality, there are several aspects of the interpretation of the results that will need to be improved before publication. These issues are discussed in the comments below. In addition, I found the manuscript long, and more importantly, unfocused and encourage the authors to revise the manuscript in order to highlight their most important findings.

**General comments**

**Introduction:** There are two sections of the introduction which should be better referenced. These sections are listed below.

- Lines 75 – 77: references are needed that describe the implementation of the new air quality improvement strategies mentioned in the text.
- Lines 84 – 93: references are needed to support the description of past air pollution issues in Lanzhou, especially for illustrating the sources of pollution as well as the meteorology discussed by the authors.

**Presentation:** Overall the quality of the writing is adequate, but copy-editing will be necessary to correct grammar and syntax errors.

**Positive matrix factorization (PMF) analyses:** Why were the highly polluted periods removed from the PMF analysis? More justification is needed to explain why this step was taken. It is concerning to see a dataset altered in such a way before a statistical analysis since that will add subjective bias. Were the highly polluted episodes removed for both the unconstrained and the constrained analyses or just for the unconstrained analyses with PMF2.exe?

In addition, the classification of the CCOA factor is not convincing. The correlation of this factor with the various tracers measured is very similar to HOA. In addition, there were no tracers measured that are specific to coal emissions for validating the assignment of this factor to coal combustion. The fact that this factor has a mass spectrum similar to HOA with a larger contribution at m/z 44 suggests that the factor may represent instead "aged" HOA.

[14]C **analysis:** Only four 24 h samples were collected. Therefore, the conclusions drawn from the [14]C analysis may not be representative and are highly dependent on the choice

of days sampled. The $^{14}$C analysis is difficult to perform, so it is understandable that only a small number of samples can be analyzed, but the manuscript must include some discussion of how aerosol properties during the four days with $^{14}$C data compare to the rest of the measurement period.

**Correlation of OOA and Odd-Oxygen ($O_X$):** There are several aspects of this analysis (on lines 800 to 822) that are seriously flawed. First, only the correlation between LV-OOA and $O_X$ has been plotted in the manuscript. It is necessary to include similar plots of SV-OOA versus $O_X$ as well as (SV-OOA + LV-OOA) versus $O_X$. Otherwise the conclusions drawn from this type of analysis regarding SOA sources are not valid. Second, it is suggested that "aromatic VOCs may be a large contributor to SOA formation". It is not clear how such a conclusion can be made based on the slope of the LV-OOA to $O_X$ plot alone, and it is impossible to draw such conclusions regarding the contributions of compounds to SOA formation without a detailed analysis of product yields and volatilities. Third, the suggestion that sources that emit modern carbon may emit more aromatic VOCs than traffic and coal combustion together is speculative. It is difficult to follow the logic behind this suggestion, but it seems to be based on the unjustified assumption that aromatic VOCs are the dominant SOA precursors at this site, which completely ignores primary semi- and intermediate volatility organic compounds (P-S/IVOCs) that are likely to be important contributors to SOA formation [e.g. *Robinson et al.*, 2007].

**Specific Comments**

**Lines 109 – 112:** This sentence is confusing the authors should re-word the sentence to make its meaning clearer.

**Line 132 – 134:** I agree that the formation and evolution mechanisms of secondary species are poorly understood, but more than one reference should be provided to support this statement. There is a large body of published work that discusses the lack of agreement between various atmospheric models and measurements of secondary aerosol species. More of this work should be cited here.

**Line 134 – 136:** More details, including references, should be given regarding the possible "fast chemical reactions" proposed in the manuscript. At the moment, these reactions are not described and seem rather mysterious.

**Line 138 – 142:** I don't agree with the phrase "the most advanced". While I do think the development of new online mass spectrometry techniques including the AMS has been very valuable, I disagree strongly with the absolute nature of this statement. One could, for example, list a variety of recently developed instruments based on optical techniques that have also been valuable.

**Lines 274 – 276:** I'm not very familiar with TEOM instruments, but it seems strange that the instrument was operated at 40°C to "minimize mass loss due to volatilization of semi-volatile aerosol compounds". This temperature is warmer than room temperature and would likely cause the evaporation of ammonium nitrate and organics.

**Lines 423 – 425:** The similar size distributions for inorganics and SOA indicate that the aerosols are internally mixed rather than externally mixed. This statement should be corrected, as well as similar statements that are made in the abstract and the conclusions.

**Lines 478 – 479:** To my eye, it appears that the concentration of $NO_2$ stays the same or even increases slightly after 14:00 rather than decreasing.

**Lines 578 – 589:** The entire discussion of BBOA aging in this paragraph is not well-supported by the measurements presented in the manuscript. It is suggested that the BBOA factor is due to oxidation of gas phase emissions. However, the measured O/C ratio for BBOA is consistent with unoxidized primary emissions [*Ortega et al.*, 2013]. In addition, the relatively large particle sizes could be due to internal mixing or coagulation of particles.

**Lines 680 – 682:** The pertinent figure for this sentence is 13b and not 13a. Also, it is not correct to say that the SV-OOA component is situated in the upper left corner. In fact, this component is positioned rather low in the two triangle diagrams.

**Lines 698 – 703:** It is difficult to understand the significance of the discussion on these lines, which seems to mostly review well-known facts regarding secondary aerosol formation. A stronger connection should be made to the OOA data or these lines should be deleted.

**Line 734:** I think this percentage should be 59%.

**Lines 736 – 740:** This sentence is very speculative. It should either be deleted or supported by measurements of actinic flux and OH radicals.

**Lines 748 – 751:** Why would emissions of COA decrease during the holiday? Presumably people still cook during the Chinese New Year.

**Line 771:** The values for SOC in the text are different from those displayed in Figure 12.

**Lines 853 – 855:** Please delete the term "VOC" from this sentence. SOA could also be formed from lower volatility P-S/IVOCs and $^{14}C$ measurements do not provide any information regarding the amount SOA specifically formed from VOCs.

**Figures 1 & 2:** I assume that the BC measurements in these figures come from the aethalometer rather than the SP2. Nevertheless, the instrument that was used should be specified in the figure caption.

**Figure 3:** It would be very interesting to include in this figure the rBC size distribution measured using the SP2.

**Figure s1a:** The text in the map is very small and difficult to read.

**References**

Ortega, A. M., D. A. Day, M. J. Cubison, W. H. Brune, D. Bon, J. A. de Gouw, and J. L. Jimenez (2013), Secondary organic aerosol formation and primary organic aerosol oxidation from biomass-burning smoke in a flow reactor during FLAME-3, *Atmos. Chem. Phys.*, *13*(22), 11551-11571, doi:10.5194/acp-13-11551-2013.

Robinson, A. L., N. M. Donahue, M. K. Shrivastava, E. A. Weitkamp, A. M. Sage, A. P. Grieshop, T. E. Lane, J. R. Pierce, and S. N. Pandis (2007), Rethinking organic aerosols: Semivolatile emissions and photochemical aging, *Science*, *315*(5816), 1259-1262, doi:10.1126/science.1133061.

---

## Author Comment (AC1) · 23 Aug 2016

The comment was uploaded in the form of a supplement:
http://www.atmos-chem-phys-discuss.net/acp-2016-278/acp-2016-278-AC1-
supplement.pdf

---

## Author Response (AR1)

The authors appreciate the two reviewers for their constructive comments and suggestions. The manuscript has been revised accordingly. Our point-by-point responses to these comments are provided below. The comments of the reviewers are printed in black italics and our responses following each comment in blue.

Response to Reviewer1

*The manuscript by Xu et al. has a comprehensive characterization of wintertime sub-micron aerosols in Lanzhou, a highly polluted city in western China. The sources of organic aerosols were investigated using positive matrix factorization and radiocarbon analysis. The results were also compared with those measured in summer. This is an important study by providing scientific data and improving our knowledge on aerosol chemistry in western China. This manuscript fits within the scope of ACP, and I recommend it for publication after addressing the following minor comments.*

*Comments:*
*1. Please consider the usage of SV-OOA and LV-OOA since this study did not measure the OA volatilities.*
 We have changed the SV-OOA and LV-OOA to LO-OOA and MO-OOA, respectively.

*2. As shown in Figure 2, POA showed an elevated contribution to PM at high mass loadings. Is this due to cooking OA, HOA or BBOA?*
Yes, the contribution of each POA factor during high mass loadings was discussed in section3.5 (Fig. 10). The results show that COA is a major contributor to the increased mass loading.

*3. Line 41-43: please show the range for the changes in fractions of OA, nitrate and sulfate.*
Done.

*4. Line 46-47: frequently calm and stagnant air conditions during wintertime in Lanzhou.*
Done.

*5. Line 51: present the O/C value during summer 2012.*
Done.

*6. Line 60: haze to air pollution events.*
Done.

*7. Line 61-62: remove "The primary OA sources were more complex during winter than during summer".*
Done.

 *8. Line 84-86: please add one or more citation.*
Done.

*9. Line 90-93: remove this sentence because of the duplication information with line 85.*
Done.

*10. Line 99: insert "recent" before "investigated:::"*
Done.

*11. Line 119: add receptor model after (CMB)*
Done.

*12. Line 122-125: add the citation.*
Done.

*13. Line 142-146: combine these two sentences into one sentence.*
Done.

*14. Line 155: remove the comma after HR-ToF-AMS and add "of" after characteristics*
Done.

*15. Line 160: change basin to valley*
Done.

*16. Line 161-164: rephrase these two sentences. Two "thus" have been used which is a little bit confused.*
Done.

*17. Line 180: add "on average" before 0.82*
Done.

*18. Line 190: RH appears at the above content.*
Done.

*19. Line 200: instruments.*
Done.

*20. Line 246: consisted to consisting.*
Done

*21. Line 270-271: where is this number (2.14) from, please add the citation.*
This sentence has been rewritten and the number has been deleted. The concentration of BC was calculated following the recommend parameters by the manufacturer.

*22. Line 275: add the dry the aerosol in the sentence.*
Done.

*23. Line 310: show the reason to remove the period of Jan.22-23.*
Please reference on the response on Reviewer #2.

*24. Line 315-317: how about seven solution results, could BBOA be separated?*
As can be seen from the following figure, we didn't find clear evidence that BBOA was separated in the seven solution (Fig. R1).

[Figure]

Fig. R1  Seven factors solution analyzed by PMF

*25. Line 373-374: there is no wind data in Figure 1, please add them.*
Done.

*26. Line 412: please show the wind data.*

Done.

*27. Line 422: primary OA to POA.*
Done.

*28. Line 443-445: please add the citation.*
Done.

*29. Line 459: change suggest to could result from*
Done

*30. Line 485-488: add "from the diurnal variation".*
Done.

*31. Line 498: is to was.*
Done.

*32. Line 502-504: Please add the diurnal variation of O/C in summer 2012.*
Done.

*33. Line 543: add "for all the PMF factor".*
Done.
*34. Line 544: change "in the past" to "in the recent".*
Done.
*35. Line 645: add "a little" before "more".*
Done.
*36. Line 653: change "have been" to "suggest".*
Done.

Response to Reviewer2

*Summary*
*This manuscript summarizes measurements of aerosol composition, concentration, and size distributions carried out during wintertime in Lanzhou, China. A large majority of the data is derived from HR-ToF-AMS measurements, although data from an aethalometer, a SP2, a SMPS, and a TEOM are also reported in the manuscript. While the data are interesting and of high quality, there are several aspects of the interpretation of the results that will need to be improved before publication. These issues are discussed in the comments below. In addition, I found the manuscript long, and more importantly, unfocused and encourage the authors to revise the manuscript in order to highlight their most important findings.*
Thank you very much for your positive comments. The manuscript is indeed a little bit long as most of AMS paper did. In this manuscript, we mainly focus on the winter-time $PM_1$ chemical composition, processes and sources in Lanzhou. The most important findings in this study include, firstly, the chemical composition and diurnal patterns of $PM_1$ species showed difference with those of summer. Secondly, the sources of organic aerosol (OA) were more complex during winter and the primary OA including HOA, BBOA, COA and CCOA was the major contributor during high air pollution. Finally, based on carbon isotopic analyses, we evaluated the contribution of fossil and modern carbon to primary and secondary organic carbon which are important to understand the source and chemical evolution of OA. In order to clarify the findings in this manuscript, we rephrase the conclusion section in the updated manuscript and emphasize on these three points.

*General comments*
*Introduction:*
*There are two sections of the introduction which should be better referenced.*
*These sections are listed below.*
*• Lines75 –77: references are needed that describe the implementation of the new air quality improvement strategies mentioned in the text.*
We add a reference in this sentence.

*• Lines 84 –93: references are needed to support the description of past air pollution issues in Lanzhou, especially for illustrating the sources of pollution as well as the meteorology discussed by the authors.*
Done.

*Presentation:*
*Overall the quality of the writing is adequate, but copy-editing will be necessary to correct grammar and syntax errors.*
*Positive matrix factorization (PMF) analyses: Why were the highly polluted periods removed from the PMF analysis? More justification is needed to explain why this step was taken. It is concerning to see a dataset altered in such a way before a statistical analysis since that will add subjective bias. Were the highly polluted episodes removed for both the unconstrained and the constrained analyses or just for the unconstrained analyses with PMF2.exe?*
The highly polluted period was removed during both constrained and unconstrained analyses, mainly based on the three points below. First, ion fitting in this period is bad. The reconstructed time series of organics during this period was offset from the measured time series and the residual is significant. Second, the mass spectra and diurnal pattern of PMF factor were influenced by this high polluted period. The mass spectra of PMF factor were more reasonable when removed this period which were more comparable with that during summer study such as HOA, COA and LO-OOA. The standard deviation of diurnal pattern on each time point of PMF factors were bigger than those after removing this period.

*In addition, the classification of the CCOA factor is not convincing. The correlation of this factor with the various tracers measured is very similar to HOA. In addition, there were no tracers measured that are specific to coal emissions for validating the assignment of this factor to coal combustion. The fact that this factor has a mass spectrum similar to HOA with a larger contribution at m/z 44 suggests that the factor may represent instead "aged" HOA.*
Response: The CCOA factor has been found in several studies in China during winter-time study (Hu et al., 2013; Elser et al., 2016; Sun et al., 2016), as in north China, coal is a major energy and heating source during wintertime that is different from developed countries. The major feature in its mass spectrum is the increased signals at m/z 91 and 115 as observed in laboratory study (Dall'Osto et al., 2013), and the high contribution at m/z 44 is related to the production of organic acids during coal combustion (Zhang et al., 2008). These spectral features were indeed observed in this study, which is similar to other observed CCOA factors in China (Hu et al., 2013; Sun et al., 2016). In addition, although the mass spectra of HOA and CCOA are similar below m/z 120, they are significant different above m/z 120, and many of them are found to be PAH-related ions (Hu et al., 2013; Sun et al., 2016).

*$^{14}$C analysis: Only four 24 h samples were collected. Therefore, the conclusions drawn from the $^{14}$C analysis may not be representative and are highly dependent on the choice of days sampled. The $^{14}$C analysis is difficult to perform, so it is understandable that only a small number of samples can be analyzed, but the manuscript must include some discussion of how aerosol properties during the four days with $^{14}$C data compare to the rest of the measurement period.*

Response: The sentence below has been added in the updated manuscript (section 2.3.3): "Here, we use the results of these four filter samples to represent the average situation of the field sampling. During the field study period, the air mass and aerosol source are pretty stable which mainly originated from regional sources as illustrated from the consistent variations of chemical composition (section 3.1.3). This can also be evidenced from the relative calm meteorological conditions during the whole sampling period (section 3.1.1)."

*Correlation of OOA and Odd-Oxygen (OX): There are several aspects of this analysis (on lines 800 to 822) that are seriously flawed. First, only the correlation between LV-OOA and OX has been plotted in the manuscript. It is necessary to include similar plots of SV-OOA versus OX as well as (SV-OOA + LV-OOA) versus OX. Otherwise the conclusions drawn from this type of analysis regarding SOA sources are not valid. Second, it is suggested that "aromatic VOCs may be a large contributor to SOA formation". It is not clear how such a conclusion can be made based on the slope of the LV-OOA to OX plot alone, and it is impossible to draw such conclusions regarding the contributions of compounds to SOA formation without a detailed analysis of product yields and volatilities. Third, the suggestion that sources that emit modern carbon may emit more aromatic VOCs than traffic and coal combustion together is speculative. It is difficult to follow the logic behind this suggestion, but it seems to be based on the unjustified assumption that aromatic VOCs are the dominant SOA precursors at this site, which completely ignores primary semi- and intermediate volatility organic compounds (P-S/IVOCs) that are likely to be important contributors to SOA formation [e.g. Robinson et al., 2007].*

Response: We did not include LO-OOA (named SV-OOA in the original manuscript) in the correlation of OOA and Ox owing to the different synchronization of LO-OOA and Ox (Fig. R2). It seems LO-OOA varied two to three hours earlier than Ox possibly due to other origination for LO-OOA such as down mixing of mixing-layer aerosol, which is a popular phenomenon in the mountain-valley city (Chen et al., 2009). We add this content in the updated manuscript (section 3.7). MO-OOA was suggested to be mainly from photochemical processes under the low RH and air temperature conditions. The different slope between OOA and Ox had been suggested to result from the different VOCs precursor in the photochemical process. In the manuscript, we didn't consider the volatility of VOCs which P-S/IVOCs can also be aromatic. This conclusion is indeed too strong only based on this slope. For the origination of modern SOC, we remove these contents in the updated manuscript.

[Figure]

Fig. R2 The scatter plots for MO-OOA, LO-OOA, and (LO-OOA + MO-OOA) versus Ox.

Specific Comments
*Lines 109 –112: This sentence is confusing the authors should re-word the sentence to make its meaning clearer.*
Response: The sentence has been revised.

*Line 132 –134: I agree that the formation and evolution mechanisms of secondary species are poorly understood, but more than one reference should be provided to support this statement. There is a large*

body of published work that discusses the lack of agreement between various atmospheric models and measurements of secondary aerosol species. More of this work should be cited here.

Response: Thanks. The sentence has been revised to "However, the formation and evolution mechanisms of those secondary species were poorly understood which previous models tended to underestimate the secondary species budget in polluted regions (e.g., Volkamer et al., 2006)".

Line 134 –136: More details, including references, should be given regarding the possible "fast chemical reactions" proposed in the manuscript. At the moment, these reactions are not described and seem rather mysterious.

Response: This sentence has been deleted.

Line 138 –142: I don't agree with the phrase "the most advanced". While I do think the development of new online mass spectrometry techniques including the AMS has been very valuable, I disagree strongly with the absolute nature of this statement. One could, for example, list a variety of recently developed instruments based on optical techniques that have also been valuable.

Response: Agree. The sentence has been revised to "…appear to be advance on probing…".

Lines 274 –276: I'm not very familiar with TEOM instruments, but it seems strange that the instrument was operated at 40°C to "minimize mass loss due to volatilization of semi-volatile aerosol compounds". This temperature is warmer than room temperature and would likely cause the evaporation of ammonium nitrate and organics.

Response: The TEOM was normally operated at 50°C to remove the water vapor. In this study, we changed the inlet temperature to 40°C to minimize the loss of semi-volatile species. The sentence has been revised to "The TEOM was operated at a temperature of 40 °C other than normal operation condition (50 °C) in order to minimize mass loss due to volatilization of semi-volatile aerosol compounds".

Lines 423 –425: The similar size distributions for inorganics and SOA indicate that the aerosols are internally mixed rather than externally mixed. This statement should be corrected, as well as similar statements that are made in the abstract and the conclusions.

Response: Revised.

Lines 478 –479: To my eye, it appears that the concentration of NO2 stays the same or even increases slightly after 14:00 rather than decreasing.

Response: This sentence has been revised to "$NO_2$ increased from 10:00 which formed from NO consumed by OH radical and slightly decreased from 14:00 to 18:00 corresponding to the formation of nitrate and $O_3$ during afternoon."

Lines 578 –589: The entire discussion of BBOA aging in this paragraph is not well-supported by the measurements presented in the manuscript. It is suggested that the BBOA factor is due to oxidation of gas phase emissions. However, the measured O/C ratio for BBOA is consistent with unoxidized primary emissions [Ortega et al., 2013]. In addition, the relatively large particle sizes could be due to internal mixing or coagulation of particles.

Response: Thanks for your suggestion. The content of BBOA aging has been deleted and the updated manuscript only focused on the primary feature of BBOA.

*Lines 680–682: The pertinent figure for this sentence is 13b and not 13a. Also, it is not correct to say that the SV-OOA component is situated in the upper left corner. In fact, this component is positioned rather low in the two triangle diagrams.*
Response: Revised.

*Lines 698 –703: It is difficult to understand the significance of the discussion on these lines, which seems to mostly review well-known facts regarding secondary aerosol formation. A stronger connection should be made to the OOA data or these lines should be deleted.*
Response: The sentence has been deleted.

*Line 734: I think this percentage should be 59%.*
Response: Revised.

*Lines 736–740: This sentence is very speculative. It should either be deleted or supported by measurements of actinic flux and OH radicals.*
Response: The speculative part of the sentence has been deleted.

*Lines 748 –751: Why would emissions of COA decrease during the holiday? Presumably people still cook during the Chinese New Year.*
Response: During the Chinese New Year holiday (up to four weeks), many people (about half of the population of Lanzhou) would leave for their hometown, and most of the restaurants in the city were closed. For example, the students' canteen in Lanzhou University had been closed. A sentence has been added in this part to explain this reason.

*Line 771: The values for SOC in the text are different from those displayed in Figure 12.*
Response: The values in the text have been revised.

*Lines 853 –855: Please delete the term "VOC" from this sentence. SOA could also be formed from lower volatility P-S/IVOCs and 14C measurements do not provide any information regarding the amount SOA specifically formed from VOCs.*
Response: Done.

*Figures1 & 2: I assume that the BC measurements in these figures come from the aethalometer rather than the SP2. Nevertheless, the instrument that was used should be specified in the figure caption.*
Response: The information for instrument of BC has been added in the captions.

*Figure 3: It would be very interesting to include in this figure the rBC size distribution measured using the SP2.*
Response: Unfortunately, we did not process the data of SP2.

*Figure s1a: The text in the map is very small and difficult to read.*
Response: The size of text has been revised.

**Reference:**

Chen, Y., Zhao, C., Zhang, Q., Deng, Z., Huang, M., and Ma, X.: Aircraft study of Mountain Chimney Effect of Beijing, China, J. Geophys. Res., 114, D08306, 10.1029/2008JD010610, 2009.

Dall'Osto, M., Ovadnevaite, J., Ceburnis, D., Martin, D., Healy, R. M., O'Connor, I. P., Kourtchev, I., Sodeau, J. R., Wenger, J. C., and O'Dowd, C.: Characterization of Urban Aerosol in Cork City (Ireland) Using Aerosol Mass Spectrometry, Atmos Chem Phys, 13, 4997-5015, 10.5194/acp-13-4997-2013, 2013.

Elser, M., Huang, R. J., Wolf, R., Slowik, J. G., Wang, Q., Canonaco, F., Li, G., Bozzetti, C., Daellenbach, K. R., Huang, Y., Zhang, R., Li, Z., Cao, J., Baltensperger, U., El-Haddad, I., and Prévôt, A. S. H.: New Insights into Pm2.5 Chemical Composition and Sources in Two Major Cities in China During Extreme Haze Events Using Aerosol Mass Spectrometry, Atmos Chem Phys, 16, 3207-3225, 10.5194/acp-16-3207-2016, 2016.

Hu, W. W., Hu, M., Yuan, B., Jimenez, J. L., Tang, Q., Peng, J. F., Hu, W., Shao, M., Wang, M., Zeng, L. M., Wu, Y. S., Gong, Z. H., Huang, X. F., and He, L. Y.: Insights on Organic Aerosol Aging and the Influence of Coal Combustion at a Regional Receptor Site of Central Eastern China, Atmos Chem Phys, 13, 10095-10112, 10.5194/acp-13-10095-2013, 2013.

Sun, Y., Du, W., Fu, P., Wang, Q., Li, J., Ge, X., Zhang, Q., Zhu, C., Ren, L., Xu, W., Zhao, J., Han, T., Worsnop, D. R., and Wang, Z.: Primary and Secondary Aerosols in Beijing in Winter: Sources, Variations and Processes, Atmos Chem Phys, 16, 8309-8329, 10.5194/acp-16-8309-2016, 2016.

Zhang, Y., Schauer, J. J., Zhang, Y., Zeng, L., Wei, Y., Liu, Y., and Shao, M.: Characteristics of Particulate Carbon Emissions from Real-World Chinese Coal Combustion, Environmental Science & Technology, 42, 5068-5073, 10.1021/es7022576, 2008.

---

## Editor Decision (ED1)

Dear Authors:

After careful consideration of your revised manuscript, I think that several of the points brought up by the referees must be addressed more thoroughly prior to publication. Please consider my comments below. Thank you.

Eleanor Browne
* * *
Major comments:

Referee 2 expressed concerns regarding the removal of highly polluted periods prior to PMF analysis. I agree that this needs to be addressed in greater detail in the manuscript – it appears to me that nothing has been changed. Additionally, I do not understand the authors' reply to this comment. Mainly, I do not understand what is meant by "ion fitting in this period is bad." With high signals, it is typically easier to do ion fitting. Do the authors mean that new ions appear? If so, those should be fitted and addressed. The discussion on PMF results with and without this time period should at least be included in the supplementary material.

Referee 2 also expressed concerns regarding the conclusions drawn from the $^{14}$C analysis and the representativeness of the samples. I think that the authors should reconsider their interpretation of these results. In particular, I am hesitant to agree with the statement that the "…air mass and aerosol source are pretty stable…" when applied to the $^{14}$C measurements. One of the $^{14}$C measurements (January 23) was performed on a day that was removed from the data analysis due to "highly polluted" conditions, and another (January 3) was during a period before the start of the AMS measurements. Given that 50% of the filters are from time periods where the AMS data is not even considered in the manuscript, how much can we really learn from this analysis and is this analysis even appropriate?

The comment from Referee 2 regarding the CCOA factor also needs to be addressed further in the manuscript. Given the similar time trends of CCOA and BBOA, how certain is it that *m/z* 91 and the PAH-related ions are related to CCOA and not BBOA, especially given that the BBOA spectrum is somewhat constrained? Please include discussion addressing these points in the manuscript.

Specific Comments

Line 120: "…applying thousands of individual species…" Given that vaporization and ionization in the AMS result in extensive fragmentation, the AMS does not really measure "individual species." I recommend revising since this is somewhat misleading.

Line 131-132: I think a few more references would be appropriate. Particularly some that represent the more recent advancements in models.

Line 135: "…appear to be advance…" This does not make sense. Please revise.

Line 156-157: The aerosols are also influenced by very different meteorological processes between the two seasons.

Line 219: "ionic path" This is non-standard. Please consider more standard wording.

Line 231: If a background was determined only once in the study, how was the gas-phase $CO_2$ correction determined for other times during the campaign? Using a constant value is likely not appropriate, and given the emphasis put on $CO_2^+$, this needs to be explained.

Sect. 2.2.1: How was the size measurement of the AMS calibrated? Significant time is spent discussing diameter later in the manuscript so this must be addressed.

Line 248-249: "…refractory mass of the particle quantified by detection of the main light-absorbing component is rBC." This is unclear and should be rephrased.

Line 283: What density was assumed in making the comparison with the SMPS? Is a slope of 1.48 really good? Or was no density applied?

Line 335: "pretty stable" is not very meaningful and should be quantified better. Please see general comments for other concerns regarding this section.

Line 344: Where does the factor of 1.03 come from?

Line 363: Should $f_{NF\_BBOC}$ be less than 1? That is my understanding from the wording. Please clarify.

Line 376: Some of the average values (particularly temperature) are not very convincing in support of the statement of stable conditions. I assume this is due to diurnal variation. Perhaps it would be better to present average lows and highs.

Lines 394-396: Do you mean to say that you have measured a lower limit?

Sect 3.1.4: The size distribution of chloride seems more similar to the size distribution of the organics than to the other inorganics. What are the implications of this and is it consistent with your factor analysis?

Line 460-462: I find this highly speculative since the oxidizing capacity is greatly reduced in the winter. Is the increase in $SO_2$ really enough to account for changes in sulfate given lower oxidizing conditions?

Line 479: $NO_2$ is not formed from the reaction of NO with OH. Please correct.

Line 486-489: These comments regarding the inorganic species can be constrained (at least somewhat) with the appropriate analysis. I recommend either performing this analysis, or removing some of this from the paper as the major point of the paper is the analysis of the organic aerosol.

Line 533: If the HOA and BC are thought to be mainly from the same source than why is the HOA peak in the evening small than the morning while the BC ones are about equal?

Lines 597-598: There are so many points in Fig. S10 that I cannot clearly see at all any time of day dependence. Please consider some sort of data reduction to make this clear. For Fig. S10, where do the COA and HOA lines come from and why does HOA fall on the COA line in panel B? It would also be helpful to see the plot for the $C_xH_y^+$ ions at *m/z* 55 and 57 as well.

Line 638: The CCOA is not "high and left." This was pointed about by Referee 2 as well. Please fix.

Line 660: Please correct the spelling of Jimenez.

Line 673-674: "These results indicate that the atmospheric oxidation capacity during the winter was still very strong." This statement is very vague (what precisely is meant by "very strong"?) and I think somewhat strong given that there is no exploration of the oxidative budget. Please reconsider.

Line 716-717: It is not at all clear to me that a correlation exists for POA less than 15 ug/m$^3$. At 15 ug/m$^3$ SOA varies between ~4 and 12 ug/m$^3$!

Lines 798-803: This part still seems rather speculative (as pointed out by Referee 2). I do not know what we are really learning from this discussion/comparison since there is no analysis of gas-phase organics in the present manuscript.

Lines 824-826: I encourage the authors to remember that the boundary layer explanation is still rather speculative since simultaneous measurements of the boundary layer or vertical structure do not exist. Even in Fig. S8 only early morning and evening temperature profiles are shown and no data is given for noon.

Table 2 & multiple places in the text): Please be consistent and use either r or r$^2$.

Fig 1: I find this very difficult to read due to the small size and the large amount of information. Please consider making more figures. It would also be useful to see the wind direction as a wind rose.

Fig 5: How is there 50 ppbv of NO at night and still 10 ppbv of $O_3$? I would have thought that $O_3$ would have been titrated away.

Fig. 7: Please use the standard AMS colors for the ion families. Please also identify them as ions by including "+" in each name. Also, the y-axis labels are overlapping and difficult to read.

Fig. 8: I am unsure how useful this figure is given that most (if not all) of this data already appears in Fig. 4.

Fig. 9: What is the R2 value representing? A value of 0.46 seems unrealistically high for any sort of line in panel A. Is OOA supposed to be SOA?

Fig 13: Would a van Krevelen plot be more meaningful? Also, please update LV-OOA and SV-OOA to their correct names. Also Fig. 13 is referenced in the text before figures 9-12 are referenced. Please reorder the figures to be referenced in order.

---

## Author Response (AR2)

**We thank Prof. Eleanor Browne for her very helpful, insightful and constructive comments. We have carefully considered these comments (in italic), and responded to each of them (in blue or red), as appended below:**

*Major comments:*

*MC1. Referee 2 expressed concerns regarding the removal of highly polluted periods prior to PMF analysis. I agree that this needs to be addressed in greater detail in the manuscript – it appears to me that nothing has been changed. Additionally, I do not understand the authors' reply to this comment. Mainly, I do not understand what is meant by "ion fitting in this period is bad." With high signals, it is typically easier to do ion fitting. Do the authors mean that new ions appear? If so, those should be fitted and addressed. The discussion on PMF results with and without this time period should at least be included in the supplementary material.*

We feel sorry that we didn't elaborate this issue with more details in the previous version. We now have re-visited and investigated this issue. Originally, we did remove a highly polluted short period on Jan. 22-23 from the PMF analyses, as our very preliminary trial on the PMF including this period showed the mass concentrations of the OA on this day was relatively not well reproduced compared with other periods. Thus, we didn't include it in the subsequent PMF treatment. However, we now re-visited this issue, and in fact, after careful PMF matrix and error preparation following the procedure outlined by Zhang et al. (2011) (including removal of a few outlier runs rather than all data on Jan 22-23, removal of highly noisy ions, etc.), the PMF modeling on this polluted day was improved significantly, thus after careful consideration, we think it is better to add back this period. In addition, adding back the polluted day also makes it more reasonable and reliable to compare the source apportionment from carbon isotope analysis results with the AMS-PMF resolved results.

The results of the new PMF analysis results covering the whole sampling period are overall similar with previous results but there are some changes between them, such as the element ratios of the factor mass profiles, mass contributions and time series of the PMF factors (as shown in the Figures below). For example, the O/C ratio of MO-OOA increased from 0.67 to 0.80 due to that a bit more $C_xH_y^+$ signals are apportioned in the primary factors including COA, BBOA and OOA, and the mass contribution of MO-OOA correspondingly decreased from 18% to 15% on average. The diurnal pattern of HOA also shows differences with the previous one – the new HOA pattern showed relatively high level throughout the nighttime (Fig. R1). The characteristics of other PMF factors are similar with previous results. We have listed the key figures that compare the old and new PMF results for reference below. Correspondingly, all PMF results (including figures and relevant text) in the manuscript were updated in this modified version.

[Figure]

[Figure]

[Figure]

[Figure]

[Figure]

Fig. R1 The comparison between the old and new PMF results

*MC2. Referee 2 also expressed concerns regarding the conclusions drawn from the 14C analysis and the representativeness of the samples. I think that the authors should reconsider their interpretation of these results. In particular, I am hesitant to agree with the statement that the "…air mass and aerosol source are pretty stable…" when applied to the 14C measurements. One of the 14C measurements (January 23) was performed on a day that was removed from the data analysis due to "highly polluted" conditions, and another (January 3) was during a period before the start of the AMS measurements. Given that 50% of the filters are from time periods where the AMS data is not even considered in the manuscript, how much can we really learn from this analysis and is this analysis even appropriate?*

We thanks for the comment. Indeed, as carbon isotope analysis is difficult for us to perform, we in this study aimed to make a try to combine it with the AMS-PMF results to probe more information regarding the fossil/non-fossil sources for the OA in Lanzhou. Currently, we selected four days in every week during the sampling month, trying to use this daily filter sample to represent a week average by assuming the meteorological conditions aerosol sources are relatively stable. We agree with the editor and the reviewer that the sampling schedule should be improved and it will be better if more samples were included so as to make the analysis more robust. This certainly will be the

subject of our future work. For the current analyses, the wind speed was very low (on average ranging from 0.6 to 1.1 m s$^{-1}$) and the prevailing wind direction was basically typical (north and northeast) during the field study (Fig. R2). This in some extent verifies our assumption that the daily sample are in some extent representative. In addition, we have added back the PMF analysis on Jan 23, so the analysis is improved from the previous version. In addition, the very similar carbon isotopic results (the total average of $f_{NF}$ in these four filters was 55 ± 3%) further confirm the representativeness of the samples. Nevertheless, cautions and limitation of our results should be clearly stated and open to the readers, thus, we have added a few sentences in the MS to make it clear, as following:

"Here, we use the results of these four filter samples to roughly represent the average situation of the field sampling because of the relative stable meteorological conditions (section 3.1.1) and similar aerosol sources during the field study (section 3.1.3). Due to the limitation of the small amount of filter samples, the results based on this carbon isotopic data are preliminary and comprehensive validation is an ongoing work."

[Figure]

Fig. R2 Three hourly average wind rose plot during the field study

*MC3. The comment from Referee 2 regarding the CCOA factor also needs to be addressed further in the manuscript. Given the similar time trends of CCOA and BBOA, how certain is it that m/z 91 and the PAH-related ions are related to CCOA and not BBOA, especially given that the BBOA spectrum is somewhat constrained? Please include discussion addressing these points in the manuscript.*

Because the similar conventional usage of biomass and coal for heating during cold seasons in Lanzhou, it is understandable that the BBOA and CCOA had similar temporal variations. In addition, the biomass usage is relatively in a small amount, thus by using the general PMF (unconstrained mass spectra), it is difficult to separate the BBOA and CCOA. Instead, we constrained the BBOA mass spectrum (MS) using ME2, and thus we can extract the BBOA and separated another factor that is CCOA. As shown in Fig. R3, indeed the temporal variations of BBOA and CCOA were similar ($R^2$ = 0.77), yet their

mass spectra were significantly different ($R^2$ = 0.35). As is well known, the standard MS of BBOA is usually characterized by the high signals at m/z 60 and 73, and the BBOA factor shows this feature. On the other hand, a high signal fraction of m/z 91, is a common feature of CCOA found in previous studies (Dall'Osto et al., 2013). The CCOA MS was sometimes also characterized by high signal of $CO_2^+$ due to the emission of organic acids directly from coal combustion (Zhang et al., 2008). In northern China, coal combustion had been thought to be a very important source of OA due to the large and widespread use of coal for heating and other purposes during wintertime (accounted for 60% of the energy consumption), and the CCOA factor was also found in several other Chinese cities (Elser et al., 2016; Hu et al., 2016; Sun et al., 2016). In general, our CCOA factor is highly similar to those CCOA identified earlier.

Furthermore, although both the MS of BBOA and CCOA can contain the PAHs-related ions (m/z), e.g., m/z 152, 165, 178, 189, 202, 215, 226, 239, 252, 276, 300, 326, etc (Sun et al., 2016). Due to widely and more enhanced use of coal in the northern China, coal combustion was found to be the most important source of PAH during wintertime (Okuda, et al., 2006). A study by Sun et al. (2016) also suggested that 66% of PAH was from coal combustion and only 18% was from biomass burning in Beijing. To address referee's concerns, in this section we have added the following sentences to explain our PMF results:

"Note that although the similar temporal variations between BBOA and CCOA, the significant differences between their MS (in particular, m/z 91) suggested their different origins. In addition, high PAH signals had been observed in the CCOA MS, and this is consistent with previous results that the coal combustion could be a dominate source of PAHs in China (Okuda et al., 2006; Sun et al., 2016). "

[Figure]

Fig. R3 The comparisons of MS and time series between BBOA and CCOA

***Specific Comments***

*SC1. Line 120: "…applying thousands of individual species…" Given that vaporization and ionization in the AMS result in extensive fragmentation, the AMS does not really measure "individual species." I recommend revising since this is somewhat misleading.*

We have reworded this sentence as following:
"Source apportionment techniques, such as the positive matrix factorization (PMF) allow us to use thousands of fragment ions for source identification and use the real measurement uncertainties to constrain the fitting, and would thus appear more suitable to identify and apportion PM to their sources".

*SC2. Line 131-132: I think a few more references would be appropriate. Particularly some that represents the more recent advancements in models.*

We have added two more references in this sentence as following:
"However, the formation and evolution mechanisms of those secondary species were poorly understood, and previous models tended to underestimate the secondary species budget in polluted regions (e.g., Volkamer et al., 2006; Carlton et al., 2010; Hodzic et al., 2016)."

Carlton, A. G., Bhave, P. V., Napelenok, S. L., Edney, E. O., Sarwar, G., Pinder, R. W., Pouliot, G. A., and Houyoux, M.: Model Representation of Secondary Organic Aerosol in Cmaqv4.7, Environ. Sci. Technol., 44, 8553-8560, doi:10.1021/es100636q, 2010.

Hodzic, A., Kasibhatla, P. S., Jo, D. S., Cappa, C. D., Jimenez, J. L., Madronich, S., and Park, R. J.: Rethinking the Global Secondary Organic Aerosol (Soa) Budget: Stronger Production, Faster Removal, Shorter Lifetime, Atmos. Chem. Phys., 16, 7917-7941, doi:10.5194/acp-16-7917-2016, 2016.

Volkamer, R., Jimenez, J. L., San Martini, F., Dzepina, K., Zhang, Q., Salcedo, D., Molina, L. T., Worsnop, D. R., and Molina, M. J.: Secondary organic aerosol formation from anthropogenic air pollution: Rapid and higher than expected, Geophys. Res. Lett., 33, L17811, doi:10.1029/2006GL026899, 2006.

*SC3. Line 135: "…appear to be advance…" This does not make sense. Please revise.*

We have reworded this sentence as following:
"Online instruments based on mass spectrometric techniques, such as Aerodyne aerosol mass spectrometer (AMS) (Jayne et al., 2000), has advantages on probing the fast aerosol chemical processes because that the instrument can output data with a large amount of chemical information and its fine time resolution (in minutes) and mass sensitivity (in ng m$^{-3}$) (Canagaratna et al., 2007)."

*SC4. Line 156-157: The aerosols are also influenced by very different meteorological processes between the two seasons.*

Revised as suggested.

"Thus aerosols are influenced largely by very different meteorological conditions and chemical processes between the two seasons."

*SC5. Line 219: "ionic path" This is non-standard. Please consider more standard wording.*

We have reworded this sentence as following:
"The mass spectrometer in the detection section works in two modes based on the shape of the ion path, i.e., V-mode and W-mode, with high sensitivity and high chemical resolution (~6000 m/$\Delta$m), respectively."

*SC6. Line 231: If a background was determined only once in the study, how was the gas-phase CO2 correction determined for other times during the campaign? Using a constant value is likely not appropriate, and given the emphasis put on CO2+, this needs to be explained.*

We agree with the referee that conducting frequent filtered air measurements during the period of field study is useful and we will try to do so in the future. For AMS study, it is better to apply on-line measurement of atmospheric $CO_2$ concentrations for gas-phase $CO_2$ correction (Collier et al., 2013); however, we do not have $CO_2$ measurement data nearby our site. We have estimated the uncertainty of this artifact for organic-$CO_2$ in previous study (Xu et al., 2014). For a range of 350–500 ppm of gas phase $CO_2$, the organic-equivalent concentration of $CO_2^+$ is in the range of 0.22–0.31 µg m$^{-3}$. Based on our filtered air measurement, we estimate an average $CO_2$ concentration of 400 ppm during this study, which corresponds to ~ 0.25 org-eq µg m$^{-3}$ of $CO_2^+$. This value was incorporated in the fragmentation table and subtracted from measured $CO_2^+$ signal to determine OrgCO$_2$ (or Org44). Given that the average OrgCO$_2$ of this study is ~ 1.7 org-eq µg m$^{-3}$, we estimate that applying a constant gas phase $CO_2$ subtraction may introduce ~ −2 – 4% uncertainty in $fCO_2^+$, which is very small. This point is now mentioned in the updated manuscript as following:

"Note that since no in-situ measurement of gas phase $CO_2$, the subtraction of a constant $CO_2$ signal (400 ppm based on filtered-air measurement in this study) may introduce uncertainties in the quantification of the organic-$CO_2^+$ signal. However, this artifact was expected to be small (less than 5% error in organic-$CO_2^+$ quantification) due to the high OA concentration (Xu et al., 2014)."

*SC7. Sect. 2.2.1: How was the size measurement of the AMS calibrated? Significant time is spent discussing diameter later in the manuscript so this must be addressed.*

The size calibration was performed following the general protocol used in the AMS community. We used standard polystyrene latex (PSL) spheres (Duke Scientific Corp., Palo Alto, CA) (100-700nm) and mono-dispersed ammonium nitrate particles (100-300nm), respectively. This information was now added in the updated manuscript:

"The instrument was calibrated for ionization efficiency (IE), inlet flow rate, and particle sizes using the standard procedure described by (Jayne et al., 2000). For example, the size calibration was performed following the general protocol used in the AMS community. We used standard polystyrene latex (PSL) spheres (Duke Scientific Corp., Palo Alto, CA) (100-700nm) and mono-dispersed ammonium nitrate particles (100-300nm), respectively. "

SC8. Line 248-249: "…refractory mass of the particle quantified by detection of the main light-absorbing component is rBC." This is unclear and should be rephrased.

We have reworded this sentence as following:
"The SP2 uses an intra-cavity Nd:YAG laser at 1064 nm to determine the light scattering and laser-induced incandescence of individual rBC (namely material associated with a strongly absorbing component at 1064 nm)."

SC9. Line 283: What density was assumed in making the comparison with the SMPS? Is a slope of 1.48 really good? Or was no density applied?

Sorry that we didn't make it clear here. We did not apply the density in the comparison between AMS and SMPS data. The slope (1.48) is the scatter plot between AMS mass concentrations and SMPS-determined volumes (assuming spherical particles). This slope can generally be used to represent the average density of the $PM_1$ and is comparable with other studies. The detail discussion is in the section 3.1.2.

We have reworded this sentence as following:
"This CE value was further validated by the consistency and reasonable slope between HR-ToF-AMS measured mass concentrations and SMPS-determined particle volumes (section 3.1.2, $R^2$ = 0.9, slope = 1.48)."

SC10. Line 335: "pretty stable" is not very meaningful and should be quantified better. Please see general comments for other concerns regarding this section.

We have reworded this sentence as response in MC2.

SC11. Line 344: Where does the factor of 1.03 come from?

A reference has been added as following:
"Zhang, Y. L., Huang, R. J., El Haddad, I., Ho, K. F., Cao, J. J., Han, Y., Zotter, P., Bozzetti, C., Daellenbach, K. R., Canonaco, F., Slowik, J. G., Salazar, G., Schwikowski, M., Schnelle-Kreis, J., Abbaszade, G., Zimmermann, R., Baltensperger, U., Prévôt, A. S. H., and Szidat, S.: Fossil vs. Non-fossil sources of fine carbonaceous aerosols in four chinese cities during the extreme winter haze episode of 2013, Atmos. Chem. Phys., 15, 1299-1312, 10.5194/acp-15-1299-2015, 2015b."

*SC12. Line 363: Should fNF_BBOC be less than 1? That is my understanding from the wording. Please clarify.*

Thanks. The original sentence was used to describe the possible source of biomass burning from soft coal, but we could not find the exact ratio of this kind of burning emission. So we assume the contribution of this soft coal to non-fossil fuel carbon was negligible .

This sentence has been revised as following:
"BBOC is estimated to be originated from biomass burning, i.e., $f_{NF\_BBOC}$ = 1;"

*SC13. Line 376: Some of the average values (particularly temperature) are not very convincing in support of the statement of stable conditions. I assume this is due to diurnal variation. Perhaps it would be better to present average lows and highs.*

Agree, the description for meteorological conditions have been changed to the average diurnal variation as following:

"The measurement site mainly received air masses from northern and northeastern directions associated with low wind speeds (WS) ranging from 0.6 to 1.1 m s$^{-1}$ (on daily average: 0.8 ± 0.2 m s$^{-1}$). The mountains to the north and south of the city could significantly reduce the wind speeds. Air temperature ranged from −5.0 to 6.6 °C (average = 0.6 ± 3.9 °C) for the diurnal variation during the campaign, but had an evident increase after the Chinese New Year (January 31, 2014) (Fig. 1a). No precipitation event occurred during the campaign, and RH was pretty low ranging from 16.8 to 39.5% (on daily average = 27.5 ± 7.4%) for the diurnal variation. Overall, the meteorological conditions during the campaign were much stable and dryer than those during summer 2012 (on average: 1.2 ± 0.6 m s$^{-1}$ for WS and 60 ± 17 % for RH)."

*SC14. Lines 394-396: Do you mean to say that you have measured a lower limit?*

Yes, this value represents a lower limit.

*SC15. Sect 3.1.4: The size distribution of chloride seems more similar to the size distribution of the organics than to the other inorganics. What are the implications of this and is it consistent with your factor analysis?*

Due to lack of other data, we don't know exactly the reason why the consistent size distribution between chloride and organics. All the fragments of chloride show similar size distribution (Fig. R4). One possible reason may be related with organochlorine which could be emitted from coal combustion and trash burning. The temporal variations of chloride in this study indeed showed good correlations with BBOA, CCOA, and LO-OOA ($R^2$, 0.45 – 0.52).

We have added a sentence in this section for explaining this phenomenon:
"Note that chloride also showed a wider distribution which was more similar with organics other than sulphate and nitrate. This was not observed during 2012 summer and could be related with OA emitted from coal combustion and biomass burning during wintertime."

[Figure]

Fig. R4 The size distribution chloride and its corresponding fragments

*SC16. Line 460-462: I find this highly speculative since the oxidizing capacity is greatly reduced in the winter. Is the increase in SO2 really enough to account for changes in sulfate given lower oxidizing conditions?*

We agree that the oxidizing capacity is greatly reduced during the wintertime, which has recently aroused widely concerns such as WINTER field campaign in USA. Although, the production of OH from ozone photolysis is reduced by more than an order of magnitude, recent studies have shown that nitryl chloride (ClNO2), together with nitrous acid (HONO), can be an important source of OH radicals in the wintertime (Young et al., 2012). In addition, the increased aerosol volume/surface is another potential factor contributing the sulfate formation during wintertime in Beijing, China (Zheng et al., 2015). The production rate of sulfate through heterogeneous reactions can be estimated by

$$\frac{dC_{S(IV)}}{dt} \approx k[S(IV)(aq)] \cdot [oxidants\ (aq)] \cdot V_{aerosol}$$

In which $C_{S(IV)}$ is the sulfate concentration, $k$ is the effective rate coefficient, $[S(IV)(aq)]$ is the $S(IV)$ concentration in the aqueous phase of aerosol, $[oxidants\ (aq)]$ is the concentration of oxidants in the aqueous phase of aerosols, and $V_{aerosol}$ is the volume concentration of humidified aerosol at ambient.

We have reworded this sentence in the updated manuscript as following:

"The significantly higher concentration of sulphate during winter than summer could result from a higher amount of precursor $SO_2$ emission, wintertime hydroxyl radical formation, and the increased aerosol particle surface due to high PM loadings that facilitated the heterogonous conversion of $SO_2$ to sulphate in winter (Yong et al., 2012; Puaede et al., 2015; Zheng et al., 2015)."

Young, C. J., Washenfelder, R. A., Roberts, J. M., Mielke, L. H., Osthoff, H. D., Tsai, C., Pikelnaya, O., Stutz, J., Veres, P. R., Cochran, A. K., VandenBoer, T. C., Flynn, J., Grossberg, N., Haman, C. L., Lefer, B., Stark, H., Graus, M., de Grouw, J., Gilman, J. B., Kuster, W. C., and Brown, S. S.: Vertically resolved measurements of nighttime radical reservoirs in Los Angeles and their contribution to the urban radical budget, Environ. Sci. Technol., 46, 10965–10973, doi:10.1021/es302206a, 2012.

Pusede, S. E., VandenBoer, T. C., Murphy, J. G., Markovic, M. Z., Young, C. J., Veres, P. R., Roberts, J. M., Washenfelder, R. A., Brown, S. S., Ren, X., Tsai, C., Stutz, J., Brune, W. H., Browne, E. C., Wooldridge, P. J., Graham, A. R., Weber, R., Goldstein, A. H., Dusanter, S., Griffith, S. M., Stevens, P. S., Lefer, B. L., and Cohen, R. C.: An Atmospheric Constraint on the No2 Dependence of Daytime near-Surface Nitrous Acid (Hono), Environ. Sci.Technol., 49, 12774-12781, doi:10.1021/acs.est.5b02511, 2015.

Zheng, G. J., Duan, F. K., Su, H., Ma, Y. L., Cheng, Y., Zheng, B., Zhang, Q., Huang, T., Kimot, T., Chang, D., Poschl, U., Cheng, Y. F., and He, K. B.: Exploring the severe winter haze in Beijing: the impact of synoptic weather, regional transport and heterogeneous reactions. Atmos. Chem. Phys., 15(6), 2969-2983 doi:10.5194/acp-15-2969-2015, 2015.

*SC17. Line 479: NO2 is not formed from the reaction of NO with OH. Please correct.*

We have reworded this sentence as following:
"$NO_2$ increased from 10:00 which formed from NO consumed by $O_3$ and slightly decreased from 14:00 to 18:00 corresponding to the photolysis of $NO_2$ and the formation of nitric acid during afternoon."

*SC18. Line 486-489: These comments regarding the inorganic species can be constrained (at least somewhat) with the appropriate analysis. I recommend either performing this analysis, or removing some of this from the paper as the major point of the paper is the analysis of the organic aerosol.*

After careful consideration, we have removed this part.

*SC19. Line 533: If the HOA and BC are thought to be mainly from the same source than why is the HOA peak in the evening small than the morning while the BC ones are about equal?*

We agree that BC was not only from traffic emission. The source of BC could be originated and/or associated with different sources such as biomass burning, coal

combustion, traffic emission etc. In our previous study (Xu et al., 2014), we found ~50% of BC was associated with traffic emission during 2012 summer. During wintertime in Lanzhou, the temporal variation of BC had tight correlation with HOA ($R^2$ = 0.71), BBOA ($R^2$ = 0.56), and CCOA ($R^2$ = 0.56). The higher evening peak of BC could be related with the higher evening peak of BBOA and CCOA. In the updated manuscript, we emphasis the morning and evening peaks of BC were contributed from multiple sources.

*SC20. Lines 597-598: There are so many points in Fig. S10 that I cannot clearly see at all any time of day dependence. Please consider some sort of data reduction to make this clear. For Fig. S10, where do the COA and HOA lines come from and why does HOA fall on the COA line in panel B? It would also be helpful to see the plot for the CxHy+ ions at m/z 55 and 57 as well.*

Fig. S10 has been revised to be clearly. We change the $C_xH_yO^+$ ions plot to the $C_xH_y^+$ ions plot which is used to diagnose the PMF results of COA and HOA. The reference for HOA and COA lines has been added.

*SC21. Line 638: The CCOA is not "high and left." This was pointed about by Referee 2 as well. Please fix.*

We have reworded this sentence as following:
"The CCOA also locates in a lower left position in the triangle plot defined by Ng et al. (2010) (Fig. 9a)."

*SC22. Line 660: Please correct the spelling of Jimenez.*

Done.

*SC23. Line 673-674: "These results indicate that the atmospheric oxidation capacity during the winter was still very strong." This statement is very vague (what precisely is meant by "very strong"?) and I think somewhat strong given that there is no exploration of the oxidative budget. Please reconsider.*

We have reworded this sentence as following:
"These results indicate that the atmospheric oxidation capacity during winter was still somewhat strong."

*SC24. Line 716-717: It is not at all clear to me that a correlation exists for POA less than 15 ug/m3. At 15 ug/m3 SOA varies between ~4 and 12 ug/m3!*

Sorry for that we didn't make it clear here. In this sentence, we try to emphasize the consistent variation of POA and SOA during a low PM loading period. As shown in the figure, the points were relative tightly correlated below ~15 µg m$^{-3}$, while they were very scatted above this concentration value. Because there are too many points, the mass

concentrations of points have a wide range. This sentence has been reworded as following:
"It is clear that POA and SOA show relative tight correlation during the periods of POA less than ~15 µg m$^{-3}$ associated with low mass fractions of OA."

SC25. Lines 798-803: This part still seems rather speculative (as pointed out by Referee 2). I do not know what we are really learning from this discussion/comparison since there is no analysis of gas-phase organics in the present manuscript.

In this part, we try to get insight of the formation mechanism of SOA based on the results of PMF and carbon isotope analyses. Although the gas-phase organics were not measured during this study, the content in this part could shine some lights for understanding the sources of SOA which is useful for further study in this region. In this case, we have kept this part in the manuscript, but we also modified some sentences to make it clear and avoid over-interpretation.

SC26. Lines 824-826: I encourage the authors to remember that the boundary layer explanation is still rather speculative since simultaneous measurements of the boundary layer or vertical structure do not exist. Even in Fig. S8 only early morning and evening temperature profiles are shown and no data is given for noon.

We have reworded this sentence as following:
"Further analysis indicated that the first peak was resulted from the contribution of multiple combustion sources and could be related with the variations of the boundary layer heights during morning which accumulated the air pollutants from early morning and until the break-up at around noon time (such influences should be further verified in the future with simultaneous measurements from boundary layer heights).

SC27. Table 2 & multiple places in the text): Please be consistent and use either r or r2.

This has been consistent to $R^2$ in the updated manuscript.

SC28. Fig 1: I find this very difficult to read due to the small size and the large amount of information. Please consider making more figures. It would also be useful to see the wind direction as a wind rose.

Sorry for it is unclear. Because this is a combo plot which is very helpful to compare the data with each other, we think it is better to keep this figure as a whole, but we also try to adjust it to make it more clear. In addition, we add a wind rose plot in Fig. S1.

SC29. Fig 5: How is there 50 ppbv of NO at night and still 10 ppbv of O3? I would have thought that O3 would have been titrated away.

We noticed this issue, but don't know its exact reason. These data was provided and downloaded from the MEP-China station. Although the value is somewhat strange, we think the diurnal variations should be correct.

*SC30. Fig. 7: Please use the standard AMS colors for the ion families. Please also identify them as ions by including "+" in each name. Also, the y-axis labels are overlapping and difficult to read.*

We add "+" in each family ion name and the display of this figure are more clear now. We keep the color for each family ion. Although there are default colors in the PIKA panel for each family ion, different AMS group has applied different color (Hu et al., 2016). The ion colors used in this manuscript have been applied in many published AMS paper.

*SC31. Fig. 8: I am unsure how useful this figure is given that most (if not all) of this data already appears in Fig. 4.*

Thanks for your suggestion. After careful consideration, we keep this figure as the diurnal contributions of each factor in this figure are frequently used in section 3.4. The diurnal contribution of each factor is important for understanding the source variation during the different time of day.

*SC32. Fig. 9: What is the R2 value representing? A value of 0.46 seems unrealistically high for any sort of line in panel A. Is OOA supposed to be SOA?*

$R^2$ represents the correlation between POA and SOA in this figure. OOA is a surrogate of SOA, and in the updated figure, OOA has been changed to SOA. After using the new PMF results, the $R^2$ is 0.1 now.

*SC33. Fig 13: Would a van Krevelen plot be more meaningful? Also, please update LV-OOA and SV-OOA to their correct names. Also Fig. 13 is referenced in the text before figures 9-12 are referenced. Please reorder the figures to be referenced in order.*

Agree. We apply a Krevelen plot in the updated Fig. 14 and use the slope of H:C versus O:C of AMS data to explain the formation mechanisms of organic aerosol. The terms of OOA factors have been updated. In addition, the order of figures has been updated too.

**Reference**

Collier, S., and Zhang, Q.: Gas-Phase Co2 Subtraction for Improved Measurements of the Organic Aerosol Mass Concentration and Oxidation Degree by an Aerosol Mass Spectrometer, Environmental Science & Technology, 47, 14324-14331, 10.1021/es404024h, 2013.

Dall'Osto, M., Ovadnevaite, J., Ceburnis, D., Martin, D., Healy, R. M., O'Connor, I. P., Kourtchev, I., Sodeau, J. R., Wenger, J. C., and O'Dowd, C.: Characterization of Urban Aerosol in Cork City (Ireland) Using Aerosol Mass Spectrometry, Atmos. Chem. Phys., 13, 4997-5015, doi:10.5194/acp-13-4997-2013, 2013.

Elser, M., Huang, R. J., Wolf, R., Slowik, J. G., Wang, Q., Canonaco, F., Li, G., Bozzetti, C., Daellenbach, K. R., Huang, Y., Zhang, R., Li, Z., Cao, J., Baltensperger, U., El-Haddad, I., and Prévôt, A. S. H.: New Insights into Pm2.5 Chemical Composition and Sources in Two Major Cities in China During Extreme Haze Events Using Aerosol Mass Spectrometry, Atmos. Chem. Phys., 16, 3207-3225, doi:10.5194/acp-16-3207-2016, 2016.

Hu, W., Hu, M., Hu, W., Jimenez, J. L., Yuan, B., Chen, W., Wang, M., Wu, Y., Chen, C., Wang, Z., Peng, J., Zeng, L., and Shao, M.: Chemical Composition, Sources, and Aging Process of Submicron Aerosols in Beijing: Contrast between Summer and Winter, J. Geophys. Res., 121, 2015JD024020, doi:10.1002/2015JD024020, 2016.

Okuda, T., Naoi, D., Tenmoku, M., Tanaka, S., He, K., Ma, Y., Yang, F., Lei, Y., Jia, Y., and Zhang, D.: Polycyclic Aromatic Hydrocarbons (Pahs) in the Aerosol in Beijing, China, Measured by Aminopropylsilane Chemically-Bonded Stationary-Phase Column Chromatography and Hplc/Fluorescence Detection, Chemosphere, 65, 427-435, doi:10.1016/j.chemosphere.2006.01.064, 2006.

Sun, Y., Du, W., Fu, P., Wang, Q., Li, J., Ge, X., Zhang, Q., Zhu, C., Ren, L., Xu, W., Zhao, J., Han, T., Worsnop, D. R., and Wang, Z.: Primary and Secondary Aerosols in Beijing in Winter: Sources, Variations and Processes, Atmos. Chem. Phys., 16, 8309-8329, doi:10.5194/acp-16-8309-2016, 2016.

Xu, J., Zhang, Q., Chen, M., Ge, X., Ren, J., and Qin, D.: Chemical composition, sources, and processes of urban aerosols during summertime in northwest China: insights from high-resolution aerosol mass spectrometry, Atmos. Chem. Phys., 14, 12593-12611, doi:10.5194/acp-14-12593-2014, 2014.

Young, C. J., Washenfelder, R. A., Roberts, J. M., Mielke, L. H., Osthoff, H. D., Tsai, C., Pikelnaya, O., Stutz, J., Veres, P. R., Cochran, A. K., VandenBoer, T. C., Flynn, J., Grossberg, N., Haman, C. L., Lefer, B., Stark, H., Graus, M., de Grouw, J., Gilman, J. B., Kuster, W. C., and Brown, S. S.: Vertically resolved measurements of nighttime radical reservoirs in Los Angeles and their contribution to the

urban radical budget, Environ. Sci. Technol., 46, 10965–10973, doi:10.1021/es302206a, 2012.

Zhang, Q., Jimenez, J. L., Canagaratna, M. R., Ulbrich, I. M., Ng, N. L., Worsnop, D. R., and Sun, Y.: Understanding atmospheric organic aerosols via factor analysis of aerosol mass spectrometry: a review, Anal. Bioanal. Chem., 401, 3045-3067, doi:10.1007/s00216-011-5355-y, 2011.

Zhang, Y., Schauer, J. J., Zhang, Y., Zeng, L., Wei, Y., Liu, Y., and Shao, M.: Characteristics of Particulate Carbon Emissions from Real-World Chinese Coal Combustion, Environ. Sci. Technol., 42, 5068-5073, doi:10.1021/es7022576, 2008.

Zheng, G. J., Duan, F. K., Su, H., Ma, Y. L., Cheng, Y., Zheng, B., Zhang, Q., Huang, T., Kimot, T., Chang, D., Poschl, U., Cheng, Y. F., and He, K. B.: Exploring the severe winter haze in Beijing: the impact of synoptic weather, regional transport and heterogeneous reactions. Atmos. Chem. Phys., **15**(6), 2969-2983 doi:10.5194/acp-15-2969-2015, 2015.

---

## Author Response (AR3)

We appreciate Prof. Eleanor Browne for her careful review and thoughtful comments. The manuscript has been revised accordingly and the comments are responded point-by-point. The comments are in black and our response in blue or red.

Comments:

(1) Line 222: delete "chemical"

Done.

(2) Line 314-315: In the response to my earlier comments it was stated that 22-23 January was included in the PMF analysis. Please delete this sentence if that is true.

Sorry about that. This sentence has been deleted.

(3) Fig 2b: The right axis shows the % not the accumulated data number.

Thank you very much! We have reworded this sentence as following (line 1298-1299):

"The right axis in (b) shows the percentage of the data number in each bin to the total data number."

(4) Fig 5: The inconsistencies between O3 and NO (non-zero O3 at night with 50 ppb NO) need to be acknowledged in the text. This was provided in the response to earlier comments, but could be expanded upon in the text.

Agree. We add a sentence in the updated manuscript as following (line 503-506):

"Note that during night time the diurnal variation of $O_3$ still showed a background (~10 ppb) although the concentrations of NO were up to 50 ppb. This inconsistent was likely due to the instrument drift in the MEP station during long-term observation, however it seemed that the pattern and the amplitude of the diurnal variation of $O_3$ were reasonable."

(5) Line 523: Is this really a smaller peak or is there just a constant offset? It is hard to tell from the figure. It seems that an offset could be explained by background/residual aerosol being more oxidized in the summer than it is in the winter. This should be clarified in the text.

The morning peak of O/C during summer 2012 slowly increased from 8:00 am, while the peak during winter 2014 increased from ~12 pm. It is indeed that there was a significant offset between these two O/C diurnal variations and we add a sentence in the updated manuscript to mention this phenomenon as following (line 529-531):

"In addition, an offset was existed for the O/C diurnal variation between the 2012 and 2014 studies especially for night time, which suggested that background/residual aerosol in the summer were more oxidized than in the winter."

(6) Line 660-661: Were the PAH signals observed in this data set? If so, an extended mass spec showing the PAH ions and the differences between BBOA and CCOA should be included in the supplement. Similarly, how is the PAH entry in Table 2 derived. This must be clarified.

Sorry for not clear on PAH data. We obtained the PAH data using V-mode data from SQUIRREL panel based on the default fragmentation table. We add a sentence in the updated manuscript to clarify this as following (line 298-300):

"In addition, the concentration of PAH was generated in SQUIRREL panel based on the default fragmentation table (Dzepina et al., 2007)."

For PAH signals in BBOA and CCOA, because we performed PMF analysis below $m/z$ 120, no PAH signals were observed in these two spectra. In the original sentence, we tried to emphasize this phenomenon on other studies. We have rephrased this sentence as following (line 668-670):

"In addition, high PAH signals had been observed in the CCOA MS (Sun et al., 2016), and this is consistent with previous results that the coal combustion could be a dominate source of PAHs in China (Okuda et al., 2006)."

(7) Line 734-735: I find this wording very confusing and the figure hard to interpret. There are so many data points in this figure that it is unclear whether the reader can truly see the data points or if some points are hidden under others. Additionally, I believe the text means to say that when Org/PM is less than ~0.5 AND POA is < 15 ug/m3 there is a tighter correlation.

Yes, the figure is indeed a bit complex due to the overlap of data points. Nevertheless, the aim was to show that when POA is low and Org/PM$_1$ fraction is low, SOA and POA correlated relatively well, while when POA mass concentrations and Org/PM$_1$ fractions were large, the correlation became weak, so as to qualitatively represent the significant role of POA for heavy pollution events, compared to that during summertime. Also, we agree with the editor, and we have further make it clear what we mean is for the case of POA < 15 $\mu g\ m^{-3}$ and Org/PM$_1$ < 0.5, and calculated the R$^2$. The sentences have been rewritten as following (line 742-747):

"Overall, the SOA and POA correlate weakly but when POA concentrations were less than ~15 $\mu g\ m^{-3}$ and the OA/PM$_1$ mass fractions were less than 0.5 (data points with green/blue colors), they have relative tight correlation (R$^2$ = 0.2). When POA concentrations and Org/PM$_1$ fractions were large, POA and SOA show almost no

correlation, indicating the importance of POA in the severe aerosol pollutions in Lanzhou during winter."

(8) Line 764: The filter (PM2.5) and AMS (PM1) measure different sizes of aerosol. This needs to be addressed here. Is this agreement still good when this is taken into account?

We add a sentence to mention the different size cute for these two sampling methods. In order to address the reviewer's concern, we also explain the high ratio of $OC_{AMS}/OC_{filter}$ due to the negative artifact of filter samples as following (line 772-777):

"The average ratio of $OC_{AMS}/OC_{filter}$ was ~1.5 for these two filters although the smaller size cute for AMS than filter sampler (PM1 vs. PM2.5). The possible reasons were likely due to the analytical uncertainties of different instruments (30% for AMS and 20% for OCfilter), which was also observed in other studies (Zotter et al., 2014), and negative artifacts for the filter samples."

(9) Line 862-863: This number is actually quite variable (44% on 23 January and 68% on 15 January). Please revise to acknowledge this, particularly given the limited variability of the C-14 measurements.

Agree. We add the uncertainty for the average contribution of fossil and non-fossil carbon in SOC based on the standard deviation of $f_{NF}$ as following (line 788-789):

"For all AMS data, the $f_F$ and $f_{NF}$ in POC were 50% and 50%, while for SOC were 34 ± 10% and 66 ± 10%."

---

## Author Response (AR4)

Comment: Congratulations on your manuscript. Prior to publication, please make sure that the numbers for fraction non-fossil fuel for SOC are consistent in lines 789 and 873. In one location it is 66+/-10% and in the other it is 60+/-10%.

We appreciate Prof. Eleanor Browne for her carefully review.
Sorry for making mistake on this number which should be 66 ±10%. We have revised it in the updated manuscript.